# Are all outliers alike?
# On Understanding the Diversity of Outliers for Detecting OODs

## Abstract

Deep neural networks (DNNs) are known to produce incorrect predictions with very high confidence on out-of-distribution (OOD) inputs. This limitation is one of the key challenges in the adoption of deep learning models in high-assurance systems such as autonomous driving, air traffic management, and medical diagnosis. This challenge has received significant attention recently, and several techniques have been developed to detect inputs where the model's prediction cannot be trusted. These techniques use different statistical, geometric, or topological signatures. This paper presents a taxonomy of OOD outlier inputs based on their source and nature of uncertainty. We demonstrate how different existing detection approaches fail to detect certain types of outliers. We utilize these insights to develop a novel integrated detection approach that uses multiple attributes corresponding to different types of outliers. Our results include experiments on CIFAR10, SVHN and MNIST as in-distribution data and Imagenet, LSUN, SVHN (for CIFAR10), CIFAR10 (for SVHN), KMNIST, and F-MNIST as OOD data across different DNN architectures such as ResNet34, WideResNet, DenseNet, and LeNet5.

## 1 Introduction

Deep neural networks (DNNs) have achieved remarkable performance-levels in many areas such as computer vision (Gkioxari et al., 2015), speech recognition (Hannun et al., 2014), and text analysis (Majumder et al., 2017). But their deployment in the safety-critical systems such as self-driving vehicles (Bojarski et al., 2016), aircraft collision avoidance (Julian & Kochenderfer, 2017), and medical diagnoses (De Fauw et al., 2018) is hindered by their brittleness. One major challenge is the inability of DNNs to be self-aware of when new inputs are outside the training distribution and likely to produce incorrect predictions. It has been widely reported in literature (Guo et al., 2017a; Hendrycks & Gimpel, 2016) that deep neural networks exhibit overconfident incorrect predictions on inputs which are outside the training distribution. The responsible deployment of deep neural network models in high-assurance applications necessitates detection of out-of-distribution (OOD) data so that DNNs can abstain from making decisions on those.

Recent approaches for OOD detection consider different statistical, geometric or topological signatures in data that differentiate OODs from the training distribution. For example, the changes in the softmax scores due to input perturbations and temperature scaling have been used to detect OODs (Hendrycks & Gimpel, 2016; Liang et al., 2017; Guo et al., 2017b). Papernot & McDaniel (2018) use the conformance among the labels of the nearest neighbors while Tack et al. (2020) use cosine similarity (modulated by the norm of the feature vector) to the nearest training sample for the detection of OODs. Lee et al. (2018) consider the Mahalanobis distance of an input from the in-distribution data to detect OODs. Several other metrics such as reconstruction error (An & Cho, 2015), likelihood-ratio between the in-distribution and OOD samples (Ren et al., 2019), trust scores (ratio of the distance to the nearest class different from the predicted class and the distance to the predicted class) (Jiang et al., 2018), density function (Liu et al., 2020; Hendrycks et al., 2019a), probability distribution of the softmax scores (Lee et al., 2017; Hendrycks et al., 2019b; Tack et al., 2020; Hendrycks et al., 2019a) have also been used to detect OODs. All these methods attempt to develop a uniform approach with a single signature to detect all OODs accompanied by empirical

evaluations that use datasets such as CIFAR10 as in-distribution data and other datasets such as SVHN as OOD.

Our study shows that OODs can be of diverse types with different defining characteristics. Consequently, an integrated approach that takes into account the diversity of these outliers is needed for effective OOD detection. We make the following three contributions in this paper:

- **Taxonomy of OODs.** We define a taxonomy of OOD samples that classify OODs into different types based on aleatoric vs epistemic uncertainty (Hüllermeier & Waegeman, 2019), distance from the predicted class vs the distance from the tied training distribution, and uncertainty in the principal components vs uncertainty in non-principal components with low variance.

- **Incompleteness of existing uniform OOD detection approaches.** We examine the limitations of the state-of-the-art approaches to detect various types of OOD samples. We observe that not all outliers are alike and existing approaches fail to detect particular types of OODs. We use a toy dataset comprising two halfmoons as two different classes to demonstrate these limitations.

- **An integrated OOD detection approach.** We propose an integrated approach that can detect different types of OOD inputs. We demonstrate the effectiveness of our approach on several benchmarks, and compare against state-of-the-art OOD detection approaches such as the ODIN (Liang et al., 2017) and Mahalanobis distance method (Lee et al., 2018).

## 2 OOD TAXONOMY AND EXISTING DETECTION METHODS

DNNs predict the class of a new input based on the classification boundaries learned from the samples of the training distribution. Aleatory uncertainty is high for inputs which are close to the classification boundaries, and epistemic uncertainty is high when the input is far from the learned distributions of all classes (Hora, 1996; Hüllermeier & Waegeman, 2019). Given the predicted class of a DNN model on a given input, we can observe the distance of the input from the distribution of this particular class and identify it as an OOD if this distance is high. We use this top-down inference approach to detect this type of OODs which are characterized by an inconsistency in model's prediction and input's distance from the distribution of the predicted class. Further, typical inputs to DNNs are high-dimensional and can be decomposed into principal and non-principal components based on the direction of high variation; this yields another dimension for classification of OODs. We, thus, categorize an OOD using the following three criteria.

1. Is the OOD associated with higher epistemic or aleatoric uncertainty, i.e., is the input away from in-distribution data or can it be confused between multiple classes?

2. Is the epistemic uncertainty of an OOD sample unconditional or is it conditioned on the class predicted by the DNN model?

3. Is the OOD an outlier due to unusually high deviation in the principal components of the data or due to small deviation in the non-principal (and hence, statistically invariant) components?

Figure 1 demonstrates different types of OODs which differ along these criteria. Type 1 OODs have high epistemic uncertainty and are away from the in-distribution data. Type 2 OODs have high epistemic uncertainty with respect to each of the 3 classes even though approximating all in-distribution (ID) data using a single Guassian distribution will miss these outliers. Type 3 OODs have high aleatoric uncertainty as they are close to the decision boundary between class 0 and class 1. Type 4 and 5 have high epistemic uncertainty with respect to their closest classes. While Type 4 OODs are far from the distribution along the principal axis, Type 5 OODs vary along a relatively invariant axis where even a small

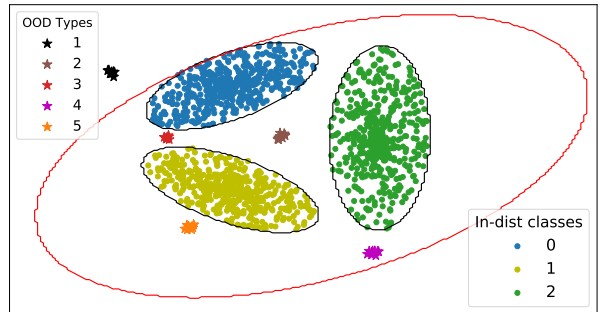

Figure 1: The different types of OODs in a 2D space with three different classes. The class distributions are represented as Gaussians with black boundaries and the tied distribution of all training data is a Gaussian with red boundary.

deviation indicates that the sample is an OOD.

**Limitations of Existing Detection Methods.** We empirically demonstrate the limitations of existing OOD detection methods on a two-dimensional (2D) half-moon dataset with two classes. As shown in Figure 2, we consider three clusters of OOD samples: cluster A (black), B (brown) and C(red). Figure 2 (right) shows the 2D penultimate features of the classifier.

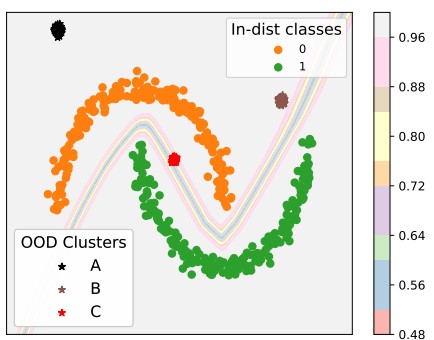 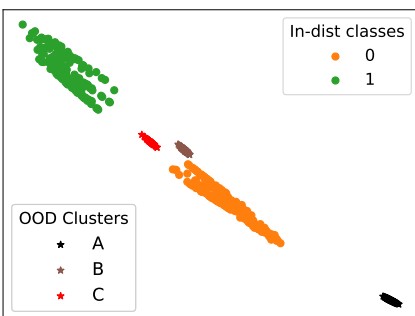

Figure 2: The cluster A (black), cluster B (brown), and cluster C (red) clusters represent the OOD due to epistemic uncertainty in the tied training distribution, epistemic uncertainty in the class-conditional training distribution, and the aleatoric uncertainty in the class-conditional distribution, respectively. (Left) shows the training data of the 2 half-moon classes and the 3 OOD clusters in the input space along with the trained classifier's boundary and its softmax scores. (Right) shows the ID samples and the OODs after projection to the 2D feature space (penultimate layer) of the DNN.

Different approaches differ in their ability to detect different OOD types as illustrated in Figure 3.

- Figure 3(a) shows that the Mahalanobis distance (Lee et al., 2018) from the mean and tied co-variance of all the training data in the feature space cannot detect OODs in the clusters B and C corresponding to class-conditional epistemic uncertainty and aleatoric uncertainty, respectively. It attains the overall true negative rate (TNR) of 39.09% at the 95% true positive rate (TPR).

- Figure 3(b) shows that the softmax prediction probability (SPB) (Hendrycks & Gimpel, 2016) cannot detect the OODs in cluster A corresponding to high epsitemic uncertainty. The TNR ( at 95% TPR) reported by the SPB technique is 60.91%.

- Figure 3(c) shows that class-wise Principal Component Analysis (PCA) (Hoffmann, 2007) cannot detect OODs in cluster C corresponding to high aleatoric uncertainty. We performed PCA of the two classes separately in the feature space and used the minimum reconstruction error to detect OODs. This obtained overall TNR of 80.91% (at 95% TPR).

- Figure 3(d) shows that K-Nearest Neighbor (kNN) (Papernot & McDaniel, 2018) non-conformance in the labels of the nearest neighbors cannot detect OODs in clusters A and B with high epistemic uncertainty. The overall TNR (at 95% TPR) reported by this technique is 15%.

These observations can be explained by the focus of different detection techniques on measuring different forms of uncertainty. This motivates our integrated OOD detection method.

## 3 INTEGRATED OOD DETECTION METHOD

Complementary information about different OOD types can be used to detect a wider range of OODs. Figure 4 shows the improvement in the TNR of the OOD detector composed with information about different classes of OODs on the two half-moons dataset. Non-conformity in the labels of the nearest neighbors captures OODs in cluster C. Mahalanobis distance from the tied in-distribution detects OODs in cluster A. Reconstruction error from the PCA of the 2 class distributions captures OODs in cluster B. Softmax scores further strengthens the OOD detection by reporting OODs in cluster C that are undetected by the other three methods.

The integrated OOD detection approach, thus, uses the following attributes, each specialized in detecting a specific type (or a combination of types) of OODs:

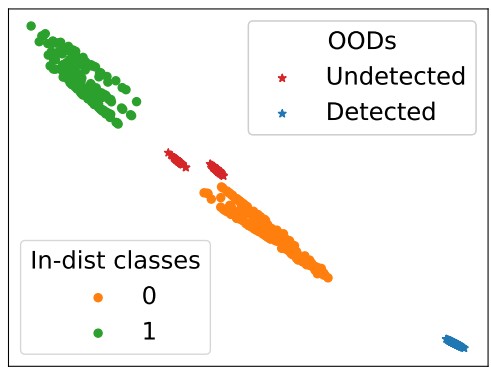
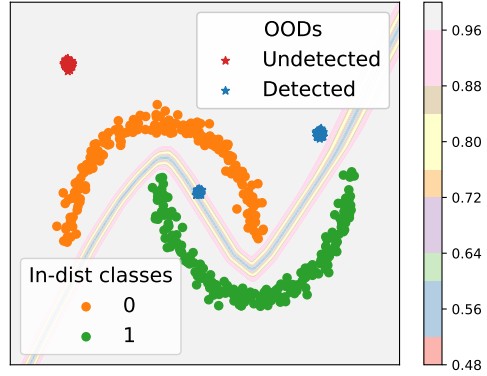
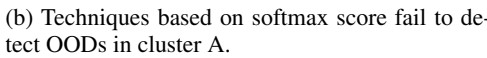

(a) Estimating distance from the tied in-distribution fails to detect OOD clusters B and C.

(b) Techniques based on softmax score fail to detect OODs in cluster A.

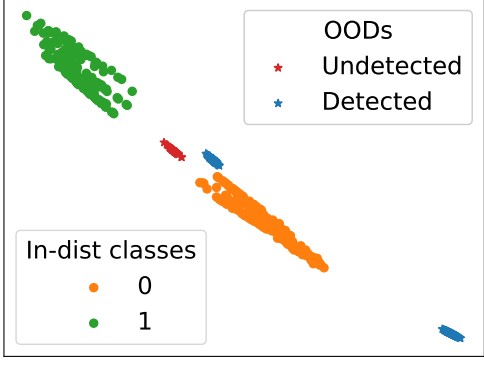
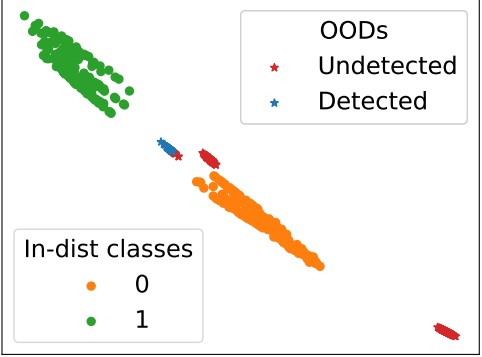

(c) Estimating distance from the class-wise in-distribution fails to detect OODs in cluster C.

(d) Non-conformance among the nearest neighbors fails to detect OODs in cluster A and B.

Figure 3: Detected OODs are shown in blue and undetected OODs are in red. Different techniques fail to detect different types of OODs.

1. Mahalanobis distance from the in-distribution density estimate that considers either tied (Lee et al., 2018) or class-wise covariance estimate. This attribute captures the overall or class-conditional epistemic uncertainty of an OOD. Our refinement to also use class-wise covariance significantly improves detection of OODs when coupled with PCA approach described below.

2. Conformance measure among the variance of the Annoy (Bernhardsson, 2018) nearest neighbors calculated as the Mahalanobis distance of the input's conformance to the closest class conformance. Our experiments found this to be very effective in capturing aleatoric uncertainty. This new attribute is a fusion of nearest-neighbor and Mahalanobis distance methods in literature.

3. Prediction confidence of the classifier as the maximum softmax score on the perturbed input where the perturbation used is the same as ODIN approach (Liang et al., 2017). This boosts the detection of high aleatoric uncertainty by sharpening the class-wise distributions.

4. Reconstruction error using top 40% of PCA components where the components are obtained via class conditional PCA of the training data. This boosts the detection of high class-wise epistemic uncertainty by eliminating irrelevant features.

This fusion of attributes from existing state-of-the-art detection methods and new attributes was found to be the most effective integrated appraoch capable of detecting the different types of OODs. We evaluated it on several benchmarks as discussed in Section 4 with ablation study in Appendix.

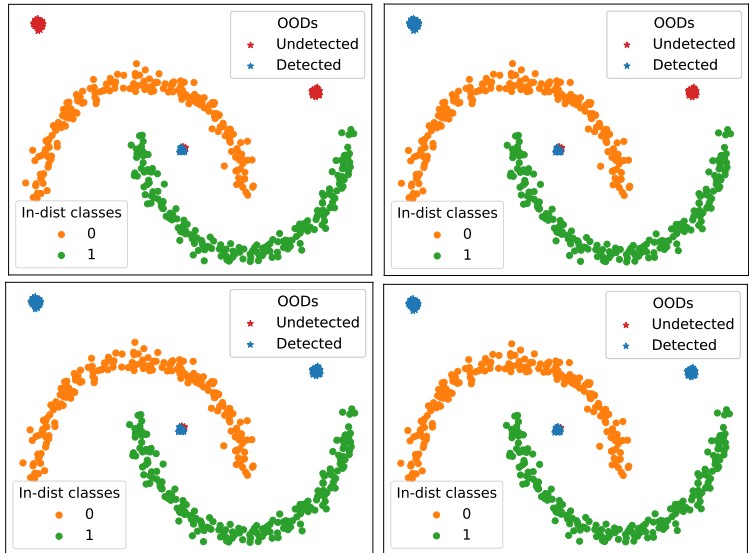

Figure 4: Complementary information about different types of OODs improves detection. (Top-left) **15%** TNR with non-conformance among the labels of the nearest neighbors. (Top-right) Adding Mahalanobis distance over the tied in-distribution improves TNR to **54.09%**. (Bottom-left) Adding Class-wise PCA further improves TNR to **95.91%** TNR. (Bottom-right) Adding softmax score further improves TNR to **99.55%**. TPR is 95% in all the cases.

## 4 EXPERIMENTAL RESULTS

**Attributes forming the signature of the OOD detector used in the experiments** The signature of the OOD detector used in the experiments is the weighted sum of four attributes, one from each of the following four categories:

1. **Distance from the in-distribution density estimate:** We use mahalanobis distance of the input with respect to the closest class conditional distribution. The parameters of this distance are chosen from one of the following two categories:
   - empirical class means and tied empirical covariance of training samples
   - empirical class means and empirical class covariance of training samples

2. **Reconstruction error:** We perform class conditional PCA empirically from the training samples. We use minimum reconstruction error of the input from the top 40% eigen vectors of the class conditional eigen spaces.

3. **Prediction confidence of the classifier:** We use maximum value of the temperature scaled softmax scores ($S$) on the perturbed input. Perturbations to the input ($x$) are made according to the following equation (Liang et al., 2017)

$$\widetilde{x} = x - \epsilon\text{sign}(-\nabla_x \log S_{\hat{y}(x;T)}) \tag{1}$$

The values of the magnitude of noise ($\epsilon$) and the temperature scaling parameter ($T$) are chosen from one of the following three categories:
   - $\epsilon = 0$ and $T = 0$
   - $\epsilon = 0$ and $T = 10$
   - $\epsilon = 0.005$ and $T = 10$

4. **Conformance measure among the nearest neighbors:** We compute an m-dimensional feature vector to capture the conformance among the input's nearest neighbors in the training samples, where m is the dimension of the input. We call this m-dimensional feature vector as the conformance vector. The conformance vector is calculated by taking the mean deviation along each dimension of the nearest neighbors from the input. We hypothesize that this deviation for the in-distribution samples would vary from the OODs due to aleatory uncertainty.

The value of the conformance measure is calculated by computing mahalanobis distance of the input's conformance vector to the closest class conformance distribution. Similar to the distance for the in-distribution density estimate, the parameters of this mahalanobis distance are chosen from the following two categories:

- empirical class means and tied empirical covariance on the conformance vectors of the training samples
- empirical class means and empirical class covariance on the conformance vectors of the training samples

The value of the number of the nearest neighbors is chosen from the set $\{10, 20, 30, 40, 50\}$ via validation. We used Annoy (Approximate Nearest Neighbors Oh Yeah) (Bernhardsson, 2018) to compute the nearest neighbors.

The weights of the four attributes forming the signature of the OOD detector are generated in the following manner. We use a small subset (1000 samples) of both the in-distribution and the generated OOD data to train a binary classifier using the logistic loss. The OOD data used to train the classifier is generated by perturbing the in-distribution data using the Fast Gradient Sign attack (FGSM) (Goodfellow et al., 2014). The trained classifier (or OOD detector) is then evaluated on the real OOD dataset at the True Positive Rate of 95%. The best result, in terms of the highest TNR on the validation dataset (from the training phase of the OOD detector), from the twelve combinations of the aforementioned sub-categories (one from each of the four attributes) are then reported on the test (or real) OOD datasets.

**Datasets and metrics.** We evaluate the proposed integrated OOD detection on benchmarks such as CIFAR10 (Krizhevsky et al., 2009) and SVHN (Netzer et al., 2011). We consider standard metrics (Hendrycks & Gimpel, 2016; Liang et al., 2017; Lee et al., 2018) such as the true negative rate (TNR) at 95% true positive rate (TPR), the area under the receiver operating characteristic curve (AUROC), area under precision recall curve (AUPR), and the detection accuracy (DTACC) to evaluate our performance.

**DNN-based classifier architectures.** To demonstrate that the proposed approach generalizes across various network architectures, we consider a wide range of DNN models such as , ResNet (He et al., 2016), WideResNet (Zagoruyko & Komodakis, 2016), and DenseNet (Huang et al., 2017).

**Comparison with the state-of-the-art.** We compare our approach with the three state-of-the-art approaches: SPB (Hendrycks & Gimpel, 2016), ODIN (Liang et al., 2017), and Mahalanobis (Lee et al., 2018). For the ODIN method, the perturbation noise is chosen from the set $\{0, 0.0005, 0.001, 0.0014, 0.002, 0.0024, 0.005, 0.01, 0.05, 0.1, 0.2\}$, and the temperature $T$ is chosen from the set $\{1, 10, 100, 1000\}$. These values are chosen from the validation set of the adversarial samples of the in-distribution data generated by the FGSM attack. For the Mahalanobis method, we consider their best results obtained after feature ensemble and input preprocessing with the hyperparameters of their OOD detector tuned on the in-distribution and adversarial samples generated by the FGSM attack. The magnitude of the noise used in pre-processing of the inputs is chosen from the set $\{0.0, 0.01, 0.005, 0.002, 0.0014, 0.001, 0.0005\}$.

**CIFAR10.** With CIFAR10 as in-distribution, we consider SVHN (Netzer et al., 2011), Tiny-Imagenet (Deng et al., 2009), and LSUN (Yu et al., 2015) as the OOD datasets. For CIFAR10, we consider two DNNs: ResNet50, and WideResNet. Table 1 shows the results.

**SVHN.** With SVHN as in-distribution, we consider CIFAR10, Imagenet, and LSUN and as the OOD datasets. For SVHN, we use the DenseNet classifier. Table 1 shows the results.

**Key observations.** We do not consider pre-processing of the inputs in our integrated OOD detector. Even without input pre-processing and with the exception of CIFAR10 OOD dataset for SVHN in-distribution trained on DenseNet, we could perform equally well (and even out-perform in most of the cases) as the Mahalanobis method on its best results generated after pre-processing the input.

We also consider a Subset-CIFAR100 as OODs for CIFAR10. Specifically, from the CIFAR100 classes, we select sea, road, bee, and butterfly as OODs which are visually similar to the ship, automobile, and bird classes in the CIFAR10, respectively. Thus, there can be numerous OOD samples due to aleatoric and class-conditional epistemic uncertainty which makes the OOD detection challenging. Figure 5 shows the t-SNE (Maaten & Hinton, 2008) plot of the penultimate features from

Table 1: Comparison of TNR, AUROC, DTACC, AUPR with SPB, ODIN and Mahalanobis methods

| In-dist (model) | OOD Dataset | Method | TNR | AUROC | DTACC | AUPR |
|---|---|---|---|---|---|---|
| CIFAR10 (ResNet50) | | | | | | |
| | SVHN | SPB | 44.69 | **97.31** | 86.36 | 87.78 |
| | | ODIN | 63.57 | 93.53 | 86.36 | 87.58 |
| | | Mahalanobis | 72.89 | 91.53 | 85.39 | 73.80 |
| | | Ours | **85.90** | 95.14 | **90.66** | **80.01** |
| | Imagenet | SPB | 42.06 | 90.8 | 84.36 | 92.6 |
| | | ODIN | 79.48 | 96.25 | 90.07 | **96.45** |
| | | Mahalanobis | 94.26 | **97.41** | 95.16 | 93.11 |
| | | Ours | **95.19** | 97.00 | **96.02** | 90.92 |
| | LSUN | SPB | 48.37 | 92.78 | 86.97 | 94.45 |
| | | ODIN | 87.29 | 97.77 | 92.65 | 97.96 |
| | | Mahalanobis | 98.17 | 99.38 | 97.38 | 98.69 |
| | | Ours | **99.36** | **99.65** | **98.57** | **98.96** |
| CIFAR10 (WideResNet) | | | | | | |
| | SVHN | SPB | 45.46 | 90.10 | 82.91 | 82.52 |
| | | ODIN | 57.14 | 89.30 | 81.14 | 75.48 |
| | | Mahalanobis | 85.86 | 97.21 | 91.87 | **94.69** |
| | | Ours | **88.95** | **97.61** | **92.46** | 92.84 |
| | LSUN | SPB | 52.64 | 92.89 | 86.81 | 94.13 |
| | | ODIN | 79.60 | 96.08 | 89.74 | 96.23 |
| | | Mahalanobis | 95.69 | 98.93 | 95.41 | 98.99 |
| | | Ours | **98.84** | **99.63** | **97.72** | **99.25** |
| SVHN (DenseNet) | | | | | | |
| | Imagenet | SPB | 79.79 | 94.78 | 90.21 | 97.2 |
| | | ODIN | 79.8 | 94.8 | 90.2 | 97.2 |
| | | Mahalanobis | **99.85** | **99.88** | **98.87** | **99.95** |
| | | Ours | 98.02 | 98.34 | 98.00 | 97.05 |
| | LSUN | SPB | 77.12 | 94.13 | 89.14 | 96.96 |
| | | ODIN | 77.1 | 94.1 | 89.1 | 97.0 |
| | | Mahalanobis | **99.99** | **99.91** | **99.23** | **99.97** |
| | | Ours | 99.74 | 99.79 | 99.08 | 99.65 |
| | CIFAR10 | SPB | 69.31 | 91.9 | 86.61 | 95.7 |
| | | ODIN | 69.3 | 91.9 | 86.6 | 95.7 |
| | | Mahalanobis | **97.03** | **98.92** | **96.11** | **99.61** |
| | | Ours | 94.87 | 98.41 | 94.97 | 98.76 |

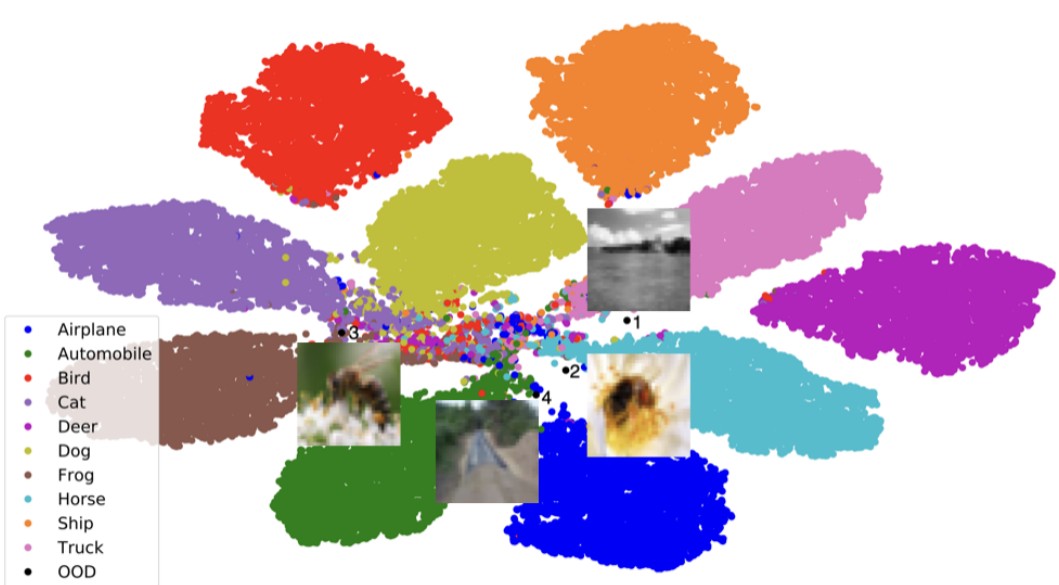

Figure 5: t-SNE plot of the penultimate layer feature space of ResNet50 trained on CIFAR10. We show four OOD images from the SCIFAR100. OOD 1 and OOD 2 are far from the distributions of all classes and thus represent OODs due to epistemic uncertainty. OOD 3 and OOD 4 are OODs due to aleatoric uncertainty as they lie closer to two class distributions. Third OOD is closer to the cat and frog classes of the ID and forth OOD is closer to the airplane and automobile classes of the ID. Mahalanobis distance cannot detect these OODs but our integrated approach can detect them.

the ResNet50 model trained on CIFAR10. We show 4 examples of OODs (2 due to epistemic and 2 due to aleatoric uncertainty) from Subset-CIFAR100. These OODs were detected by our integrated approach but missed by the Mahalanobis approach.

These observations justify the effectiveness of integrating multiple attributes to detect OOD samples.

**Additional experimental results in the appendix.** We also compare the performance of the integrated OOD detector with the SPB, ODIN and Mahalanobis detector in supervised settings, as reported by the Mahalanobis method for OOD detection (Lee et al., 2018). These results include experiments on CIFAR10, SVHN and MNIST as in-distribution data and Imagenet,LSUN, SVHN (for CIFAR10), CIFAR10 (for SVHN), KMNIST, and F-MNISTas OOD data across different DNN architectures such as ResNet34, WideResNet,DenseNet, and LeNet5. All these results, along with the ablation studies on OOD detectors with single attributes are included in the Appendix. In almost all of the reported results in the Appendix, our OOD detector could outperform the compared state-of-the-art methods with improvements of even 2X higher TNR at 95% TPR in some cases.

## 5 DISCUSSION AND FUTURE WORK

Recent techniques propose refinement in the training process of the classifiers for OOD detection. Some of these techniques include fine-tuning the classifier's training with an auxiliary cost function for OOD detection (Hendrycks et al., 2019a; Liu et al., 2020). Other techniques make use of self-supervised models for OOD detection (Tack et al., 2020; Hendrycks et al., 2019b). We perform preliminary experiments to compare the performance of these techniques with our integrated OOD detector that makes use of the feature space of the pre-trained classifiers to distinguish in-distribution samples from OODs. Our approach does not require modification of the training cost function of the original task. These results are reported in the Appendix. We consider making use of the feature space of in our OOD detection technique as a promising prospective future work. Another direction of the future work is to explore the score functions used in these refined training processes for OOD detection (Liu et al., 2020; Hendrycks et al., 2019a; Tack et al., 2020; Hendrycks et al., 2019b) as attributes (or categories of attributes) forming the signature of the integrated OOD detector. Another

avenue of future work is to explore OOD generation techniques other than adversarial examples generated by the FGSM attack for training of the integrated OOD detector.

## 6 CONCLUSION

We introduced a taxonomy of OODs and proposed an integrated approach to detect different types of OODs. Our taxonomy classifies OOD on the nature of their uncertainty and we demonstrated that no single state-of-the-art approach detects all these OOD types. Motivated by this observation, we formulated an integrated approach that fuses multiple attributes to target different types of OODs. We have performed extensive experiments on a synthetic dataset and several benchmark datasets (e.g., MNIST, CIFAR10, SVHN). Our experiments show that our approach can accurately detect various types of OODs coming from a wide range of OOD datasets such as KMNIST, Fashion-MNIST, SVHN, LSUN, and Imagenet. We have shown that our approach generalizes over multiple DNN architectures and performs robustly when the OOD samples are similar to in-distribution data.

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

# A APPENDIX

## A.1 DEFINING OODS DUE TO EPISTEMIC AND ALEATORIC UNCERTAINTY

In general, let there be $k$ classes $c_1, c_2, \ldots, c_k$ and the distribution of training data for each class is $p(x|c_i)$. The overall training distribution is denoted by $p(x)$. Now, given a new input $\hat{x}$ to the trained DNN model $M$, let $\hat{c} = M(\hat{x})$ denote the predicted class. The flowchart in Figure 6 shows different sources of uncertainty that could make $\hat{x}$ an OOD.

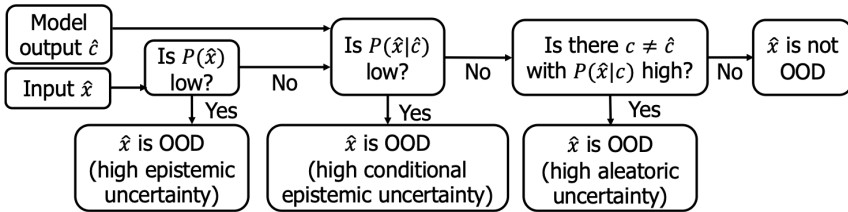

Figure 6: OODs due to High Epistemic and Aleatoric Uncertainty

## A.2 ADDITIONAL EXPERIMENTAL RESULTS

We first present preliminary results for comparison with the OOD detection techniques based on fine-tuning of the classifiers (Hendrycks et al., 2019a; Liu et al., 2020; Tack et al., 2020; Hendrycks et al., 2019b). We then present our results on various vision datasets and different architectures of the pre-trained DNN based classifiers for these datasets in comparison to the ODIN, the Mahalanobis and the SPB methods in supervised settings. Finally, we then report results from the ablation study on OOD detection with individual attributes and compare it with our integrated approach on OOD detection.

### A.2.1 COMPARISON WITH THE OOD DETECTION TECHNIQUES BASED ON REFINEMENT OF THE TRAINING PROCESS FOR CLASSIFIERS

Recent techniques propose refinement in the training process of the classifiers for OOD detection. Some of these techniques include fine-tuning the training of classifiers with a trainable cost function for OOD detection (Hendrycks et al., 2019a; Liu et al., 2020), self-supervised training of the classifiers to enhance OOD detection (Tack et al., 2020; Hendrycks et al., 2019b) etc.

We perform preliminary experiments to compare the performance of these techniques with our integrated OOD detector that uses features of the pre-trained classifiers to distinguish in-distribution samples from OODs. Table 2 compares TNR (at 95% TPR), AUROC and AUPR for the energy based OOD detector (Liu et al., 2020) and our integrated OOD detector on CIFAR10 with pre-trained WideResNet model. The integrated OOD detector was trained on in-distribution and adversarial samples generated by FGSM attack. Table 3 compares the results of the WideResNet model trained on CIFAR10 and fine-tuned with the outlier exposure from the 80 Million Tiny Images with our OOD detector that uses features from the pre-trained WideResNet model trained on CIFAR10. Since 80 Million Tiny Images dataset is no longer available for use, we used a small subset of ImageNet (treated as OOD dataset for CIFAR10 and SVHN datasets (Lee et al., 2018)) for generating OODs for training of the integrated OOD detector. Table 4 compares the OOD detection performance of the self-supervised training based OOD detector with our method. We trained our OOD detector with the in-distribution CIFAR10 as in-distribution samples and adversarial samples generated by FGSM attack from the test dataset of CIFAR10 as OODs. The trained OOD detector was then tested on LSUN as OODs and the results are reported in Table 4. With ResNet-50 as the classifier for CIFAR10, we trained our OOD detector with the in-distribution CIFAR10 as in-distribution samples and adversarial samples generated by FGSM from the test dataset of CIFAR10 as OODs. The trained OOD detector was then tested on SVHN as OODs and these results are compared with the contrastive based learning for OOD detection (Tack et al., 2020) in table 5.

Table 2: Results with Energy based OOD detector (Liu et al., 2020) / Our method.

| In-dist (model) | OOD dataset | TNR (TPR=95%) | AUROC | AUPR |
|---|---|---|---|---|
| CIFAR10 (WideResNet) | SVHN | 64.41 / **88.95** | 90.96 / **97.61** | 97.64 / **92.84** |
| | LSUN | 72.42 / **98.84** | 94.24 / **99.63** | 98.67 / **99.25** |

Table 3: Results with Outlier Exposure based OOD detector (Hendrycks et al., 2019a) / Our method.

| In-dist (model) | OOD dataset | TNR (TPR=95%) | AUROC | AUPR |
|---|---|---|---|---|
| CIFAR10 (WideResNet) | SVHN | **95.64** / 92.53 | **98.63** / 98.56 | **99.74** / 96.62 |

Table 4: Results with self-supervised learning based OOD detector (Hendrycks et al., 2019b) / Our method.

| In-dist (model) | OOD dataset | TNR (TPR=95%) | AUROC | DTACC |
|---|---|---|---|---|
| CIFAR10 (WideResNet) | LSUN | 71.3 / **98.84** | 93.2 / **99.63** | 71.0 / **97.72** |

Table 5: Results with contrastive learning based OOD detector (Tack et al., 2020) / Our method.

| In-dist (model) | OOD dataset | TNR (TPR=95%) | AUROC | DTACC | AUPR |
|---|---|---|---|---|---|
| CIFAR10 (ResNet50) | SVHN | **97.2** / 82.88 | **99.5** /96.98 | **96.7** / 91.74 | **99.6** / 94.71 |

### A.2.2 COMPARISON WITH THE STATE-OF-THE-ART OOD DETECTION METHODS IN SUPERVISED SETTINGS ON PRE-TRAINED CLASSIFIERS

We compare our results with the state-of-the-art methods in supervised settings, as reported by the Mahalanobis method for OOD detection (Lee et al., 2018). In supervised settings, a small subset of the real OOD dataset is used in the training of the OOD detector.

**Datasets and metrics.** We evaluate the proposed integrated OOD detection on benchmarks such as MNIST (LeCun et al., 1998), CIFAR10 (Krizhevsky et al., 2009), and SVHN (Netzer et al., 2011). We consider standard metrics (Hendrycks & Gimpel, 2016; Liang et al., 2017; Lee et al., 2018) such as the true negative rate (TNR) at 95% true positive rate (TPR), the area under the receiver operating characteristic curve (AUROC), area under precision recall curve (AUPR) with both in-distribution and OODs as positive samples (AUPR IN and AUPR OUT respectively), and the detection accuracy (DTACC) to evaluate our performance.

**DNN-based classifier architectures.** To demonstrate that the proposed approach generalizes across various network architectures, we consider a wide range of DNN models such as Lenet (LeCun et al., 1998), ResNet (He et al., 2016), and DenseNet (Huang et al., 2017).

**Comparison with the state-of-the-art.** We compare our approach with the three state-of-the-art approaches: SPB (Hendrycks & Gimpel, 2016), ODIN (Liang et al., 2017) and Mahalanobis (Lee et al., 2018). Since, these experiments are performed in supervised settings, we fix $T = 10$ and $\epsilon = 0.005$ for generating results from the ODIN method. For Mahalanobis distance, we consider the distance in the penultimate layer feature space as well as features from all the layers of the DNN without preprocessing of the input in either settings.

**MNIST.** With MNIST as in-distribution, we consider KMNIST (Clanuwat et al., 2018) and Fashion-MNIST(F-MNIST) (Xiao et al., 2017) as OOD datasets. For MNIST, we use the LeNet5 (LeCun et al., 1998) DNN. Results in terms of TNR (at 95% TPR), AUROC, and DTACC are reported in tables 6, 7, and 8. Table 6 shows the results with the features from the penultimate layer in comparison to the ODIN and Mahalanobis methods. Table 7 shows the results with the features from all the layers in comparison to the Mahalanobis method. Table 8 shows the results with the features from the penultimate layer in comparison to the SPB method. In all these settings, our approach outperforms the state-of-the-art approaches for both the OOD datasets. Results in comparison to AUPR IN and AUPR OUT are shown in table 9. Here also, our technique out-performs all the three OOD detectors on all the test cases.

**CIFAR10.** With CIFAR10 as in-distribution, we consider STL10 (Coates et al., 2011), SVHN (Netzer et al., 2011), Imagenet (Deng et al., 2009), LSUN (Yu et al., 2015), and a subset of CIFAR100 (SCIFAR100) (Krizhevsky et al., 2009) as OOD datasets. For CIFAR10, we consider three DNNs: DenseNet, ResNet34, and ResNet50. Results in terms of TNR (at 95% TPR), AUROC, and DTACC are reported in tables 6, 7, and 8. Table 6 shows the results with the features from the penultimate layer in comparison to the ODIN and Mahalanobis methods. Table 7 shows the results with the features from all the layers in comparison to the Mahalanobis method. Table 8 shows the results with the features from the penultimate layer in comparison to the SPB method. Results in comparison to AUPR IN and AUPR OUT are shown in tables 10, 11, and 12. Here also, the integrated OOD detection technique could out-perform the other three detectors on most of the test cases.

Note that images from STL10 and the subset of CIFAR100 are quite similar to CIFAR10 images. Furthermore, from the CIFAR100 classes, we select sea, road, bee, and butterfly as OODs which are visually similar to the ship, automobile, and bird classes in the CIFAR10, respectively.

**SVHN.** With SVHN as in-distribution, we consider STL10, CIFAR10, Imagenet, LSUN and, SCIFAR100 as OOD datasets. For SVHN, we consider two DNNs: DenseNet and ResNet34. Results in terms of TNR (at 95% TPR), AUROC, and DTACC are reported in tables 6, 7, and 8. Table 6 shows the results with the features from the penultimate layer in comparison to the ODIN and Mahalanobis methods. Table 7 shows the results with the features from all the layers in comparison to the Mahalanobis method. Table 8 shows the results with the features from the penultimate layer in comparison to the SPB method. Results in comparison to AUPR IN and AUPR OUT are shown in tables 13, and 14. Here also, the integrated OOD detection technique could out-perform the other three detectors on most of the test cases.

**Key observations.** As shown in Table 6 and Table 7, and Table 8, our approach outperforms the state-of-the-art on all three datasets and with various DNN architectures. On CIFAR10, in terms of the TNR metric, our approach with Resnet50 outperforms Mahalanobis by 56% when SVHN is OOD and our approach with Resnet34 outperforms ODIN by 36% when LSUN is OOD.

While considering STL10 and Subset-CIFAR100 as OODs for CIFAR10, the images from both these datasets are quite similar to CIFAR-10 images. Thus, there can be numerous OOD samples due to aleatoric and class-conditional epistemic uncertainty which makes detection challenging. Although our performance is low on the STL10 dataset, it still outperforms the state-of-the-art. For instance, the proposed approach achieves a 27% better TNR score than the Mahalanobis using ResNet50. On SVHN, in terms of the TNR metric, our approach outperforms ODIN and Mahalanobis by 63% and 13%, respectively on SCIFAR100 using ResNet34. The above observations justify the effectiveness of integrating multiple attributes to detect OOD samples.

### A.2.3    ABLATION STUDY

We report ablation study on OOD detection with individual attributes and compare it with our integrated approach on the penultimate feature space of the classifier in the supervised settings as described in the previous section. We call the OOD detector with Mahalanobis distance estimated on class mean and tied covariance (Lee et al., 2018) as Mahala-Tied. Detector based on Mahalanobis distance estimated on class mean and class covariance is referred as Mahala-Class. Similarly conformance among the K-nearest neighbors (KNN) measured by Mahala-Tied and Mahala-Class is referred as KNN-Tied and KNN-Class respectively in these experiments. Results for this study on CIFAR10 with DenseNet architecture, SVHN with DenseNet and ResNet34 architectures are shown in Tables 15, 16 and 17 respectively.

The integrated approach could out-perform all the single attribute based OOD detector in all the tested cases due to detection of diverse OODs. An important observation made from these experiments is that the performance of the single attribute based methods could depend on the architecture of the classifier. For example, while the performance of PCA was really bad in case of DenseNet (for both CIFAR10 and SVHN) as compared to all other methods, it could out-perform all but the integrated approach for SVHN on ResNet34.

Table 6: Results with ODIN/Mahalanobis/Our method. The best results are highlighted.

| In-dist (model) | OOD dataset | TNR (TPR=95%) | AUROC | DTACC |
|---|---|---|---|---|
| MNIST (LeNet5) | KMNIST | 67.72 / 80.52 / **91.82** | 92.98 / 96.53 / **98.3** | 85.99 / 90.82 / **94.01** |
| | F-MNIST | 58.47 / 63.33 / **74.49** | 90.76 / 94.11 / **95.55** | 83.21 / 87.76 / **90.98** |
| CIFAR10 (DenseNet) | STL10 | 8.89 / 9.23 / **15.29** | 56.31 / 62.16 / **63.96** | 55.38 / 59.57 / **61.02** |
| | SVHN | 69.96 / 83.63 / **91.29** | 92.02 / 97.1 / **98.38** | 84.1 / 91.26 / **93.28** |
| | Imagenet | 61.03 / 49.33 / **77.81** | 91.4 / 90.32 / **95.98** | 83.85 / 83.08 / **89.74** |
| | LSUN | 71.89 / 46.63 / **84.34** | 94.37 / 91.18 / **97.27** | 87.72 / 84.93 / **92.1** |
| | SCIFAR100 | 35.06 / 20.33 / **38.78** | 80.18 / 80.4 / **90.58** | 72.58 / 74.15 / **85.35** |
| CIFAR10 (ResNet34) | STL10 | 10.63 / 13.9 / **17.4** | 61.56 / 66.47 / **67.52** | 59.22 / 62.75 / **63.7** |
| | SVHN | 72.85 / 53.16 / **88.2** | 93.85 / 93.85/ **97.69** | 85.4 / 89.173 / **92.14** |
| | Imagenet | 46.54 / 68.41 / **74.53** | 90.45 / 95.02 / **95.73** | 83.06 / 88.63 / **89.73** |
| | LSUN | 45.16 / 77.53 / **81.23** | 89.63 / 96.51 / **96.87** | 81.83 / 90.64 / **91.19** |
| | SCIFAR100 | 37 / 38.39 / **61.11** | 86.13 / 88.86 / **94.74** | 78.5 / 82.51 / **90.53** |
| CIFAR10 (ResNet50) | STL10 | 12.19 / 10.33 / **16** | 60.29 / 61.95 / **66.39** | 58.57 / 59.36 / **62.28** |
| | SVHN | 86.61 / 34.49/ **91.06** | 84.41 / **98.19** / 91.98 | 91.25 / 76.72 / **93.2** |
| | Imagenet | 73.23 / 29.48 / **75.96** | 94.91 / 84.3 / **95.79** | 88.23 / 77.19 / **89.26** |
| | LSUN | 80.72 / 32.18 / **81.38** | 96.51 / 87.09 / **96.93** | 90.59 / 80.07 / **91.79** |
| | SCIFAR100 | 47.44 / 21.06 / **48.33** | 86.16 / 77.42/ **92.98** | 78.69 / 71.43 / **88.27** |
| SVHN (DenseNet) | STL10 | 45.91 / 81.66 / **87.76** | 77.6 / 96.97 / **97.63** | 72.62 / 92.29 / **93.35** |
| | CIFAR10 | 37.23 / 80.82 / **86.42** | 73.14 / 96.8 / **97.37** | 68.92 / 92.27 / **92.86** |
| | Imagenet | 62.76 / 85.44 / **93.44** | 85.41 / 97.29 / **98.38** | 79.94 / 93.39 / **94.53** |
| | LSUN | 62.91 / 76.87 / **89.73** | 86.06 / 96.37 / **97.73** | 80.04 / 92.43 / **93.55** |
| | SCIFAR100 | 48.17 / 86.06 / **96.72** | 78.94 / 97.43 / **98.24** | 73.72 / 93.02 / **96.26** |
| SVHN (ResNet34) | STL10 | 35.14 / 85.3 / **90.9** | 67.05 / 97.19 / **97.76** | 66.19 / 93.41 / **94.34** |
| | CIFAR10 | 32.6 / 85.03 / **90.34** | 66.75 / 97.05 / **97.64** | 65.37 / 93.15 / **94.29** |
| | Imagenet | 41.8 / 84.46 / **89.82** | 73 / 96.95 / **97.59** | 69.84 / 93.14 / **94.32** |
| | LSUN | 35.92 / 78.38 / **85.46** | 68.6 / 96.17 / **97.09** | 66.75 / 91.98 / **93.17** |
| | SCIFAR100 | 36.67 / 86.61 / **99.61** | 68.01 / 97.3 / **98.47** | 67.26 / 93.6 / **97.36** |

Table 7: Results with Mahalanobis/Our method with feature ensemble. The best results are highlighted.

| In-dist (model) | OOD dataset | TNR (TPR=95%) | AUROC | DTACC |
|---|---|---|---|---|
| MNIST (LeNet5) | KMNIST | 96 / **98.8** | 99.19 / **99.65** | 95.56 / **97.3** |
| | F-MNIST | 99.9 / **99.98** | 99.95 / **99.96** | 98.98 / **99.17** |
| CIFAR10 (DenseNet) | STL10 | 16.44 / **22.94** | 72.4 / **75.23** | 66.69 / **69.31** |
| | SVHN | 92.4 / **98.23** | 98.41 / **99.49** | 93.97 / **97.02** |
| | Imagenet | 96.46 / **98.8** | 99.16/ **99.63** | 95.74 / **97.55** |
| | LSUN | 98.09 / **99.64** | 99.47 / **99.85** | 96.76 / **98.45** |
| | SCIFAR100 | 27.33 / **46.5** | 83.7 / **92.17** | 77.08 / **86.69** |
| CIFAR10 (ResNet34) | STL10 | 26.14 / **29.8** | 76.23 / **76.46** | 70.33 / **70.94** |
| | SVHN | 91.53 / **97.07** | 98.4 / **99.32** | 93.63 / **96.27** |
| | Imagenet | 97.09 / **98.11** | 99.47 / **99.58** | 96.31 / **96.91** |
| | LSUN | 98.67 / **99.41** | 99.71 / **99.81** | 97.56 / **98.14** |
| | SCIFAR100 | 38.89 / **62.78** | 88.8 / **94.23** | 82.14 / **90.16** |
| CIFAR10 (ResNet50) | STL10 | 26.36 / **30.83** | 73.74 / **76.73** | 67.37 / **70.4** |
| | SVHN | 84.44 / **98.59** | 96.56 / **99.65** | 90.63 / **97.43** |
| | Imagenet | 97.87 / **99.46** | 99.58 / **99.84** | 97.09 / **98.22** |
| | LSUN | 99.21 / **99.83** | 99.64 / **99.91** | **98.39** /99.21 |
| | SCIFAR100 | 29.33 / **55** | 80.26 / **91.48** | 74.51 / **86.42** |
| SVHN (DenseNet) | STL10 | 97.31 / **98.76** | 99.14 / **99.47** | 96.23 / **97.24** |
| | CIFAR10 | 96.36 / **97.64** | 98.8 / **99.16** | 95.7 / **96.34** |
| | Imagenet | **99.89** / 99.82 | 99.88 / **99.9** | 98.85 / **98.95** |
| | LSUN | **99.99** / 99.97 | **99.91** / **99.91** | **99.26** /99.18 |
| | SCIFAR100 | 99.33 / **100** | 99.53 / **99.78** | 97.89 / **98.95** |
| SVHN (ResNet34) | STL10 | 98.44 / **98.88** | 99.31 / **99.52** | 96.91 / **97.4** |
| | CIFAR10 | 98.44 / **98.88** | 99.31 / **99.52** | 96.91 / **97.4** |
| | Imagenet | 99.83 / **99.87** | 99.85 / **99.91** | **99.07** / **99.07** |
| | LSUN | 99.87 / **99.99** | 99.83 / **99.95** | 99.5 / **99.47** |
| | SCIFAR100 | 99.83 / **100** | 99.72 / **99.91** | 98.33/ **99.56** |

Table 8: Experimental results with SPB/Our method. The best results are highlighted.

| In-dist (model) | OOD dataset | TNR (TPR=95%) | AUROC | DTACC |
|---|---|---|---|---|
| MNIST (LeNet5) | KMNIST | 69.33 / **91.82** | 93.24 / **98.3** | 86.88 / **94.01** |
| | F-MNIST | 52.69 / **74.49** | 89.19 / **95.55** | 82.77 / **90.98** |
| CIFAR10 (DenseNet) | STL10 | **15.64** / 15.29 | **64.15** / 63.96 | **62.12** / 61.02 |
| | SVHN | 39.22 / **91.29** | 88.24 / **98.38** | 82.41 / **93.28** |
| | Imagenet | 40.13 / **77.81** | 89.3 / **95.98** | 82.67 / **89.74** |
| | LSUN | 48.38 / **84.34** | 92.14 / **97.27** | 86.22 / **92.1** |
| | SCIFAR100 | 34.11 / **38.78** | 85.53 / **90.58** | 79.18 / **85.35** |
| CIFAR10 (ResNet34) | STL10 | 14.9 / **17.4** | 65.88 / **67.52** | 62.85 / **63.7** |
| | SVHN | 32.47 / **88.2** | 89.88 / **97.69** | 85.06 / **92.14** |
| | Imagenet | 44.72 / **74.53** | 91.02 / **95.73** | 85.05 / **89.73** |
| | LSUN | 45.44 / **81.23** | 91.04 / **96.87** | 85.26 / **91.19** |
| | SCIFAR100 | 38.17 / **61.11** | 88.91 / **94.74** | 82.34 / **90.53** |
| CIFAR10 (ResNet50) | STL10 | 15.33 / **16** | 66.68 / **66.39** | 63.47 / **62.28** |
| | SVHN | 44.69 / **91.06** | 97.31 / **91.98** | 86.36 /**93.2** |
| | Imagenet | 42.06 / **75.96** | 90.8 /**95.79** | 84.36 / **89.26** |
| | LSUN | 48.37 / **81.38** | 92.78 / **96.93** | 86.97 / **91.79** |
| | SCIFAR100 | 36.39 / **48.33** | 89.09 / **92.98** | 83.37 / **88.27** |
| SVHN (DenseNet) | STL10 | 72.87 / **87.76** | 92.79 / **97.63** | 87.76 / **93.35** |
| | CIFAR10 | 69.31 / **86.42** | 91.9 / **97.37** | 86.61 / **92.86** |
| | Imagenet | 79.79 / **93.44** | 94.78 / **98.38** | 90.21 / **94.53** |
| | LSUN | 77.12 / **89.73** | 94.13 / **97.73** | 89.14 / **93.55** |
| | SCIFAR100 | 76.94 / **96.72** | 94.18 / **98.24** | 89.57 / **96.26** |
| SVHN (ResNet34) | STL10 | 79.57 / **99.59** | 93.84 / **99.72** | 90.83 / **98.06** |
| | CIFAR10 | 78.26 / **90.34** | 92.92 / **97.64** | 90.03 / **94.29** |
| | Imagenet | 79.02 / **89.82** | 93.51 / **97.59** | 90.44 / **94.32** |
| | LSUN | 74.29 / **85.46** | 91.58 / **97.09** | 88.96 / **93.17** |
| | SCIFAR100 | 81.28 / **99.61** | 94.62 /**98.47** | 91.48 / **97.36** |

Table 9: Experimental Results with MNIST on Lenet5 for AUPR IN and AUPR OUT. The best results are highlighted.

| OOD Dataset | Layer | Method | AUPR IN | AUPR OUT |
|---|---|---|---|---|
| | Penultimate | | | |
| KMNIST | | Baseline | 92.47 | 92.41 |
| | | ODIN | 92.65 | 92.69 |
| | | Mahalanobis | 96.69 | 96.2 |
| | | Ours | **98.48** | **98.13** |
| Fashion-MNIST | | Baseline | 87.98 | 87.89 |
| | | ODIN | 90.94 | 89.99 |
| | | Mahalanobis | 95.24 | 91.94 |
| | | Ours | **96.64** | **92.96** |
| | All | | | |
| KMNIST | | Mahalanobis | 99.22 | 99.18 |
| | | Ours | **99.67** | **99.64** |
| Fashion-MNIST | | Mahalanobis | 99.95 | 99.94 |
| | | Ours | **99.96** | **99.96** |

Table 10: Experimental Results with CIFAR10 on DenseNet for AUPR IN and AUPR OUT. The best results are highlighted.

| OOD Dataset | Layer | Method | AUPR IN | AUPR OUT |
|---|---|---|---|---|
| | Penultimate | | | |
| STL10 | | Baseline | 64.55 | **59.37** |
| | | ODIN | 60.24 | 50.17 |
| | | Mahalanobis | 65.86 | 53.87 |
| | | Ours | **66.01** | 58.5 |
| SVHN | | Baseline | 74.53 | 94.09 |
| | | ODIN | 80.49 | 97.05 |
| | | Mahalanobis | 94.13 | 98.78 |
| | | Ours | **96.26** | **99.37** |
| Imagenet | | Baseline | 90.88 | 86.74 |
| | | ODIN | 91.32 | 90.55 |
| | | Mahalanobis | 91.32 | 88.6 |
| | | Ours | **96.22** | **95.68** |
| LSUN | | Baseline | 93.68 | 89.83 |
| | | ODIN | 94.65 | 93.39 |
| | | Mahalanobis | 92.71 | 87.74 |
| | | Ours | **97.6** | **96.74** |
| Subset CIFAR100 | | Baseline | 96.65 | 50.08 |
| | | ODIN | 95.14 | 47.64 |
| | | Mahalanobis | 95.68 | 37.86 |
| | | Ours | **98.18** | **54.83** |
| | All | | | |
| STL10 | | Mahalanobis | 77.21 | 63.45 |
| | | Ours | **78.29** | **68.14** |
| SVHN | | Mahalanobis | 96.72 | 99.31 |
| | | Ours | **98.57** | **99.81** |
| Imagenet | | Mahalanobis | 99.19 | 99.13 |
| | | Ours | **99.62** | **99.54** |
| LSUN | | Mahalanobis | 99.49 | 99.45 |
| | | Ours | **99.82** | **99.85** |
| Subset CIFAR100 | | Mahalanobis | 96.58 | 42.41 |
| | | Ours | **98.53** | **59.7** |

Table 11: Experimental Results with CIFAR10 on ResNet34 for AUPR IN and AUPR OUT. The best results are highlighted.

| OOD Dataset | Layer | Method | AUIN | AUOUT |
|---|---|---|---|---|
| | Penultimate | | | |
| STL10 | | Baseline | 67.17 | 59.74 |
| | | ODIN | 64.22 | 53.83 |
| | | Mahalanobis | 68.48 | 59.47 |
| | | Ours | **68.78** | **61.52** |
| SVHN | | Baseline | 85.4 | 93.96 |
| | | ODIN | 86.46 | 97.55 |
| | | Mahalanobis | 91.19 | 96.14 |
| | | Ours | **94.7** | **99.1** |
| Imagenet | | Baseline | 92.49 | 88.4 |
| | | ODIN | 92.11 | 87.46 |
| | | Mahalanobis | 95.77 | 94.02 |
| | | Ours | **96.32** | **94.99** |
| LSUN | | Baseline | 92.45 | 88.55 |
| | | ODIN | 91.58 | 86.5 |
| | | Mahalanobis | 97.08 | 95.78 |
| | | Ours | **97.36** | **96.29** |
| Subset CIFAR100 | | Baseline | 97.77 | 55.62 |
| | | ODIN | 97.05 | 51.57 |
| | | Mahalanobis | 97.71 | 54.11 |
| | | Ours | **99.06** | **64.53** |
| | All | | | |
| STL10 | | Mahalanobis | 77.59 | 71.15 |
| | | Ours | **77.32** | **72.38** |
| SVHN | | Mahalanobis | 96.46 | 99.37 |
| | | Ours | **98.37** | **99.73** |
| Imagenet | | Mahalanobis | 99.48 | 99.48 |
| | | Ours | **99.59** | **99.58** |
| LSUN | | Mahalanobis | 99.71 | 99.71 |
| | | Ours | **99.8** | **99.82** |
| Subset CIFAR100 | | Mahalanobis | 97.75 | 52.28 |
| | | Ours | **98.74** | **65.99** |

Table 12: Experimental Results with CIFAR10 as on ResNet50 for AUPR IN and AUPR OUT. The best results are highlighted.

| OOD Dataset | Layer | Method | AUIN | AUOUT |
|---|---|---|---|---|
| | Penultimate | | | |
| STL10 | | Baseline | 67.47 | 60.83 |
| | | ODIN | 62.79 | 55.04 |
| | | Mahalanobis | 65.14 | 54.43 |
| | | Ours | **68.54** | **59.75** |
| SVHN | | Baseline | 87.78 | 95.61 |
| | | ODIN | 93.17 | 99.03 |
| | | Mahalanobis | 71.88 | 92.54 |
| | | Ours | **95.38** | **99.34** |
| Imagenet | | Baseline | 92.6 | 87.98 |
| | | ODIN | 95.16 | 94.45 |
| | | Mahalanobis | 86.14 | 80.6 |
| | | Ours | **96.22** | **95.23** |
| LSUN | | Baseline | 94.45 | 90.41 |
| | | ODIN | 96.9 | 96.01 |
| | | Mahalanobis | 89.34 | 82.87 |
| | | Ours | **97.53** | **96.03** |
| Subset CIFAR100 | | Baseline | 97.72 | 55.29 |
| | | ODIN | 96.67 | 60.62 |
| | | Mahalanobis | 94.49 | 36.12 |
| | | Ours | **98.72** | **59.3** |
| | All | | | |
| STL10 | | Mahalanobis | 75.6 | 69.32 |
| | | Ours | **77.79** | **73.22** |
| SVHN | | Mahalanobis | 91.89 | 98.58 |
| | | Ours | **99.12** | **99.86** |
| Imagenet | | Mahalanobis | 99.56 | 99.6 |
| | | Ours | **99.84** | **99.84** |
| LSUN | | Mahalanobis | **98.91** | 99.75 |
| | | Ours | 99.72 | **99.93** |
| Subset CIFAR100 | | Mahalanobis | 94.1 | 42.96 |
| | | Ours | **97.54** | **64.12** |

Table 13: Experimental Results with SVHN as on DenseNet for AUPR IN and AUPR OUT.
The best results are highlighted.

| OOD Dataset | Layer | Method | AUIN | AUOUT |
|---|---|---|---|---|
| | Penultimate | | | |
| STL10 | | Baseline | 97.01 | 82.02 |
| | | ODIN | 89.18 | 63.71 |
| | | Mahalanobis | 99.2 | 87.32 |
| | | Ours | **99.36** | **90.31** |
| CIFAR10 | | Baseline | 95.7 | 82.8 |
| | | ODIN | 84.32 | 60.32 |
| | | Mahalanobis | 98.94 | 88.91 |
| | | Ours | **99.09** | **91.59** |
| Imagenet | | Baseline | 97.2 | 88.42 |
| | | ODIN | 90.95 | 79.59 |
| | | Mahalanobis | 99.12 | 90.22 |
| | | Ours | **99.4** | **95.15** |
| LSUN | | Baseline | 96.96 | 87.44 |
| | | ODIN | 92.03 | 79.98 |
| | | Mahalanobis | 98.84 | 85.79 |
| | | Ours | **99.17** | **92.92** |
| Subset CIFAR100 | | Baseline | 99.39 | 63.21 |
| | | ODIN | 97.24 | 45.23 |
| | | Mahalanobis | 99.82 | **72.35** |
| | | Ours | **99.88** | 68.25 |
| | All | | | |
| STL10 | | Mahalanobis | 99.75 | 96.51 |
| | | Ours | **99.77** | **98.18** |
| CIFAR10 | | Mahalanobis | 99.6 | 95.39 |
| | | Ours | **99.69** | **97.21** |
| Imagenet | | Mahalanobis | **99.96** | 99.59 |
| | | Ours | **99.96** | **99.74** |
| LSUN | | Mahalanobis | **99.97** | 99.7 |
| | | Ours | 99.95 | **99.74** |
| Subset CIFAR100 | | Mahalanobis | 99.97 | 91.41 |
| | | Ours | **99.98** | **94.54** |

Table 14: Experimental Results with SVHN as on ResNet34 for AUPR IN and AUPR OUT. The best results are highlighted.

| OOD Dataset | Layer | Method | AUIN | AUOUT |
|---|---|---|---|---|
| | Penultimate | | | |
| STL10 | | Baseline | 96.63 | 84.15 |
| | | ODIN | 84.03 | 47.26 |
| | | Mahalanobis | 99.89 | 97.5 |
| | | Ours | **99.93** | **98.2** |
| CIFAR10 | | Baseline | 95.06 | 85.66 |
| | | ODIN | 80.69 | 50.49 |
| | | Mahalanobis | 99.04 | 88.62 |
| | | Ours | **99.17** | **91.17** |
| Imagenet | | Baseline | 95.68 | 86.18 |
| | | ODIN | 84.62 | 58.28 |
| | | Mahalanobis | 99 | 88.39 |
| | | Ours | **99.19** | **90.77** |
| LSUN | | Baseline | 94.19 | 83.95 |
| | | ODIN | 82.37 | 53.12 |
| | | Mahalanobis | 98.73 | 85.11 |
| | | Ours | **99.03** | **89.03** |
| Subset CIFAR100 | | Baseline | 99.35 | 64.38 |
| | | ODIN | 95.57 | 23.04 |
| | | Mahalanobis | 99.81 | 64.4 |
| | | Ours | **99.89** | **67.9** |
| | All | | | |
| STL10 | | Mahalanobis | 99.7 | 97.03 |
| | | Ours | **99.84** | **97.86** |
| CIFAR10 | | Mahalanobis | 99.7 | 97.03 |
| | | Ours | **99.84** | **97.86** |
| Imagenet | | Mahalanobis | 99.86 | 99.14 |
| | | Ours | **99.93** | **99.62** |
| LSUN | | Mahalanobis | 99.82 | 98.85 |
| | | Ours | **99.98** | **99.64** |
| Subset CIFAR100 | | Mahalanobis | 99.98 | 93.59 |
| | | Ours | **99.99** | **95.75** |

Table 15: Ablation study with CIFAR10 on DenseNet.
The best results are highlighted.

| OOD dataset | Method | TNR (TPR=95%) | AUROC | DTACC | AUPR IN | AUPR OUT |
|---|---|---|---|---|---|---|
| SVHN | Mahala-Tied | 83.63 | 97.1 | 91.26 | 94.13 | 98.78 |
| | Mahala-Class | 71.73 | 95.16 | 87.92 | 90.77 | 97.98 |
| | KNN-Tied | 84.07 | 97.18 | 91.32 | 94.2 | 98.84 |
| | KNN-Class | 77.95 | 96.19 | 89.68 | 92.06 | 98.45 |
| | SPB | 39.22 | 88.24 | 82.41 | 74.53 | 94.09 |
| | ODIN | 69.96 | 92.02 | 84.1 | 80.49 | 97.05 |
| | PCA | 2.46 | 55.89 | 56.36 | 35.42 | 74.12 |
| | Integrated(Our) | **91.29** | **98.38** | **93.28** | **96.26** | **99.37** |
| Imagenet | Mahala-Tied | 49.33 | 90.32 | 83.08 | 91.32 | 88.6 |
| | Mahala-Class | 53.11 | 92.16 | 85.3 | 93.42 | 90.29 |
| | KNN-Tied | 51.36 | 90.73 | 83.31 | 91.75 | 88.87 |
| | KNN-Class | 57.94 | 92.74 | 86.01 | 93.67 | 91.28 |
| | SPB | 40.13 | 89.3 | 82.67 | 90.88 | 86.74 |
| | ODIN | 61.03 | 91.4 | 83.85 | 91.32 | 90.55 |
| | PCA | 4.66 | 58.68 | 57.19 | 60.66 | 54.42 |
| | Integrated(Our) | **77.81** | **95.98** | **89.74** | **96.22** | **95.68** |
| LSUN | Mahala-Tied | 46.63 | 91.18 | 84.93 | 92.71 | 87.74 |
| | Mahala-Class | 58.53 | 93.82 | 88.16 | 95.15 | 91.43 |
| | KNN-Tied | 51.48 | 92.25 | 85.96 | 93.75 | 89.13 |
| | KNN-Class | 65.17 | 94.57 | 88.6 | 95.61 | 92.61 |
| | SPB | 48.38 | 92.14 | 86.22 | 93.68 | 89.83 |
| | ODIN | 71.89 | 94.37 | 87.72 | 94.65 | 93.39 |
| | PCA | 2.06 | 53.26 | 54.88 | 57.08 | 49.33 |
| | Integrated(Our) | **84.34** | **97.27** | **92.1** | **97.6** | **96.74** |

Table 16: Ablation study with SVHN on DenseNet.
The best results are highlighted.

| OOD dataset | Method | TNR (TPR=95%) | AUROC | DTACC | AUPR IN | AUPR OUT |
|---|---|---|---|---|---|---|
| CIFAR10 | Mahala-Tied | 80.82 | 96.8 | 92.27 | 98.94 | 88.91 |
| | Mahala-Class | 82.99 | 97.11 | 92.83 | 99.05 | 89.71 |
| | KNN-Tied | 69.99 | 95.58 | 90.77 | 98.52 | 84.3 |
| | KNN-Class | 74.52 | 96.01 | 91.21 | 98.64 | 85.99 |
| | SPB | 69.31 | 91.9 | 86.61 | 95.7 | 82.8 |
| | ODIN | 37.23 | 73.14 | 68.92 | 84.32 | 60.32 |
| | PCA | 5.27 | 65.82 | 64.83 | 86.62 | 33.51 |
| | Integrated(Our) | **86.6** | **97.41** | **92.88** | **99.11** | **91.76** |
| Imagenet | Mahala-Tied | 85.44 | 97.29 | 93.39 | 99.12 | 90.22 |
| | Mahala-Class | 77.66 | 96.83 | 93.17 | 98.98 | 88.59 |
| | KNN-Tied | 65.76 | 94.67 | 89.59 | 98.18 | 80.16 |
| | KNN-Class | 73.44 | 95.69 | 90.68 | 98.55 | 84.28 |
| | SPB | 79.79 | 94.78 | 90.21 | 97.2 | 88.42 |
| | ODIN | 62.76 | 85.41 | 79.94 | 90.95 | 79.59 |
| | PCA | 5.16 | 65.08 | 65.39 | 86.65 | 32.83 |
| | Integrated(Our) | **93.46** | **98.39** | **94.54** | **99.41** | **95.16** |
| LSUN | Mahala-Tied | 76.87 | 96.37 | 92.43 | 98.84 | 85.79 |
| | Mahala-Class | 69.44 | 96.05 | 92.4 | 98.74 | 84.89 |
| | KNN-Tied | 59.64 | 93.71 | 88.22 | 97.83 | 77.17 |
| | KNN-Class | 66.96 | 94.77 | 89.45 | 98.21 | 81.27 |
| | SPB | 77.12 | 94.13 | 89.14 | 96.96 | 87.44 |
| | ODIN | 62.91 | 86.06 | 80.04 | 92.03 | 79.98 |
| | PCA | 3.19 | 62.66 | 64.7 | 85.72 | 30.37 |
| | Integrated(Our) | **89.73** | **97.73** | **93.55** | **99.17** | **92.92** |

Table 17: Ablation study with SVHN on ResNet34.
The best results are highlighted.

| OOD dataset | Method | TNR (TPR=95%) | AUROC | DTACC | AUPR IN | AUPR OUT |
|---|---|---|---|---|---|---|
| SCIFAR100 | Mahala-Tied | 86.61 | 97.3 | 93.6 | 99.81 | 64.4 |
| | Mahala-Class | 88.44 | 97.7 | 94.19 | 99.84 | 69.59 |
| | KNN-Tied | 84.67 | 96.82 | 92.83 | 99.76 | 61.08 |
| | KNN-Class | 83.72 | 96.83 | 93.05 | 99.77 | 57.58 |
| | SPB | 81.28 | 94.62 | 91.48 | 99.35 | 64.38 |
| | ODIN | 36.67 | 68.01 | 67.26 | 95.57 | 23.04 |
| | PCA | 89.94 | 97.81 | 94.52 | 99.84 | 70.83 |
| | Integrated(Our) | **99.61** | **98.47** | **97.36** | **99.89** | **67.9** |
| LSUN | Mahala-Tied | 78.38 | 96.17 | 91.98 | 98.73 | 85.11 |
| | Mahala-Class | 81.51 | 96.71 | 92.44 | 98.91 | 87.63 |
| | KNN-Tied | 77.61 | 95.98 | 91.34 | 98.61 | 85.56 |
| | KNN-Class | 78.77 | 96.05 | 91.45 | 98.62 | 85.71 |
| | SPB | 74.29 | 91.58 | 88.96 | 94.19 | 83.95 |
| | ODIN | 35.92 | 68.6 | 66.75 | 82.37 | 53.12 |
| | PCA | 82.93 | 96.88 | 92.74 | 98.97 | 88.27 |
| | Integrated(Our) | **85.46** | **97.09** | **93.17** | **99.03** | **89.03** |
| CIFAR10 | Mahala-Tied | 85.03 | 97.05 | 93.15 | 99.04 | 88.62 |
| | Mahala-Class | 86.84 | 97.41 | 93.48 | 99.15 | 90.37 |
| | KNN-Tied | 82.17 | 96.65 | 92.24 | 98.87 | 87.63 |
| | KNN-Class | 83.24 | 96.73 | 92.38 | 98.9 | 87.67 |
| | SPB | 78.26 | 92.92 | 90.03 | 95.06 | 85.66 |
| | ODIN | 32.67 | 66.75 | 65.37 | 80.69 | 50.49 |
| | PCA | 88.18 | 97.55 | 93.83 | 99.2 | 90.77 |
| | Integrated(Our) | **90.34** | **97.64** | **94.29** | **99.17** | **91.17** |

