# OpenReview forum: "Are all outliers alike?  On Understanding the Diversity of Outliers for Detecting OODs"
_ICLR.cc/2021/Conference — Reject_

### Official Review · AnonReviewer2 · 2020-10-23
**Good paper but need improvement**

**Rating:** 4
**Confidence:** 5

**Review:**

This paper introduces a taxonomy of OODs and proposed an integrated approach to detect different types of OODs. Their taxonomy classifies OOD on the nature of their uncertainty and they show that no single state-of-the-art approach detects all these OOD types. Motivated by this observation, they combine multiple existing OOD detection methods to detect various types of OODs.

In general, this paper is easy to understand. But I have the following concerns:

1. Lack of discussions about some important related work. They only compare their method to ODIN and Mahalanobis methods. But there are some other OOD detection methods which also achieve state-of-the-art results, such as [1][2][3]. Could the authors compare their method to these methods?

2. In their taxonomy, they consider examples that are very close to in-distribution as OOD. I am wondering whether we should treat those examples as OOD since they are too close to the in-distribution. I think previous works like ODIN and Mahalanobis all assume that OOD inputs are far away from the in-distribution. In the experimental setup, they consider STL10 as an OOD dataset for CIFAR10. But STL10 contains CIFAR10 alike images. It is unconvincing that we should treat those images as OOD. And I think the classifier trained on CIFAR10 may have correct predictions on some of those images. Could the authors explain why we should treat those images as OOD?

3. I am wondering whether the analysis for the simple two-dimensional dataset could be applied to high-dimensional datasets. In the high-dimensional space, their conclusion about which method detects which type of OOD may not hold. Could the authors explain it?

4. In Appendix A.2.1, they mention that the best results from the twelve combinations of the aforementioned sub-categories (one from each of the four attributions) are reported. Could the authors explain how they select the best results? Do they use the test OOD data to select the best results?

5. Could the author describe how they integrate the existing state-of-the-art detection methods in detail? It is hard for me to understand what they exactly do in their proposed method.

--------- After Reading the Updated Paper ----------

Thanks for the update. After reading the revised paper, I still have some major concerns:

1. The current experiments performed are not enough to demonstrate the effectiveness of the proposed method. The old experiment results (Table 6, 7, 8) are not convincing since the authors train a binary classifier as an OOD detector using a subset of the test OOD data, which is not realizable in practice. We should assume that the test OOD data are unknown during learning the OOD detector. The new experimental results where they train the binary classifier using adversarial examples generated on in-distribution data (follow the Mahalanobis method) in Table 1 are limited. For example, on CIFAR10, they only report results for ResNet50 and WideResNet, but I also want to know the results for DenseNet (Mahalanobis method [4] performs very well on CIFAR10/SVHN using DenseNet under the same setting).

2. Some experimental details about their method are missing. The authors mention that they train 12 binary classifiers and then select the best one on the validation dataset. But they don't provide the details about the validation dataset, which is critical for their results. Based on their previous response, it seems they use a subset of test OOD data to select the best classifier, which is not allowed I think. Based on the current description of experimental settings, it is hard for me to evaluate the reported results.

3. The proposed approach needs a lot of hyper-parameters (4 attributes, 12 combinations, the weights of the binary classifier, etc) and it is unclear how to tune these hyper-parameters and how they would affect the results. The current ablation study is limited I think.

4. This paper doesn't have rigorous analysis for why integrating different attributions would improve OOD detection. I think this is an empirical paper but the experiments provided are not sufficient to demonstrate the effectiveness of the proposed method.

To clarify, I didn't agree to raise the score previously. What I said was that the previous paper needed significant revision and I could not recommend acceptance. I still have some major concerns after reading the revised paper. Thus, I keep the same rating and think the paper is not ready for publication. I hope the authors could keep improving their paper.


[1] Hendrycks, Dan, Mantas Mazeika, and Thomas Dietterich. "Deep anomaly detection with outlier exposure." arXiv preprint arXiv:1812.04606 (2018).

[2] Liu, Weitang, et al. "Energy-based Out-of-distribution Detection." arXiv preprint arXiv:2010.03759 (2020).

[3] Lakshminarayanan, Balaji, Alexander Pritzel, and Charles Blundell. "Simple and scalable predictive uncertainty estimation using deep ensembles." Advances in neural information processing systems. 2017.

[4] Lee, Kimin, et al. "A simple unified framework for detecting out-of-distribution samples and adversarial attacks." Advances in Neural Information Processing Systems. 2018.

---

> ### Author Response · Authors · 2020-11-14
> **Addressing comments of AnonReviewer2**
>
> “STL10 contains CIFAR10 alike images… Could the authors explain why we should treat STL10 as OODs for CIFAR10?”
>
> STL10 dataset is inspired byCIFAR-10 but two datasets differ in terms of the image resolution (STL10 - 96X96 and CIFAR10 - 32X32). Since STL10 is similar to the CIFAR10 dataset, it makes the OOD detection more challenging while considering STL10 as OODs for the CIFAR10. This is the reason why we selected this pair of datasets to stress test our method.
>
> “I am wondering whether the analysis for the simple two-dimensional dataset could be applied to high-dimensional datasets. In the high-dimensional space, their conclusion about which method detects which type of OOD may not hold. Could the authors explain it?”
>
> We agree that it is hard to verify whether the observations in the 2D dataset apply directly to the high-dimensional datasets. However, in fig. 5, we show the projection of the high-dimensional features in 2D using t-SNE. Please note the highlighted examples where the OOD samples associated with both epistemic and aleatoric uncertainties are missed by the Mahalanobis [4] but were detected by our approach. Also, our ablation studies (in appendix Table. 10, 11, 12) show that individual approaches (ODIN, PCA, Mahalanobis, etc) are not sufficient to detect all types of OODs and an integrated approach is necessary.
>
> “Could the authors explain how they select the best results? Do they use the test OOD data to select the best results? Could the author describe how they integrate the existing state-of-the-art detection methods in detail?”
>
> A weighted sum of the four attributes (section A.2.1 of the appendix) forms the signature of our OOD detector. The weights of these attributes are generated in the following manner. Following the standard experimental setup [4, 6], we use a small subset of both in-distribution and OOD  data to train a binary classifier using a logistic loss. The trained classifier (or OOD detector) is then evaluated on the remaining OOD samples at the True Positive Rate of 95%.
>
> The four attributes (forming the signature of the OOD detector) are 1) distance from the in-distribution density estimate, 2) reconstruction error from the principal component analysis (PCA), 3) prediction confidence of the classifier, and 4) conformance measure among the nearest neighbors. These attributes can be computed in different ways. In our experiments, we consider 2 ways of generating distance from the in-distribution density estimate, 1 way of generating reconstruction error from PCA, 3 ways of generating prediction confidence of the classifier, and 2 ways of generating conformance measure among the nearest neighbors, resulting in a total of 12 ways of combining these four attributes (2*1*3*2) (section A.2.1 of the appendix). Out of these 12 combinations, we report the best empirical result on the test OOD data.
>
> “Lack of discussion on related work. Could the authors compare their method to these methods...”
>
> We are looking into the suggested papers to compare our results with these papers. We will provide an update on this once we have the results.
>
> [4] Lee, Kimin, et al. "A simple unified framework for detecting out-of-distribution samples and adversarial attacks." Advances in Neural Information Processing Systems. 2018.
>
> [6] Lee, Kimin, et al. "Training confidence-calibrated classifiers for detecting out-of-distribution samples." arXiv preprint arXiv:1711.09325 (2017).

---

> > ### Comment · AnonReviewer2 · 2020-11-14
> > **Concerns still remain**
> >
> > 1. I think it is not reasonable to treat STL10 images as OOD just because their resolution is different from CIFAR10. When we feed the images to the model, we need to first resize the images to the same resolution. I think the authors don't answer my question: why we should treat those images that are very similar to CIFAR10 images as OOD? What's their definition of OOD? Usually, people define OOD as inputs with new classes (see open-world classification and previous related work about OOD detection).
> > 2. I still think that in the high-dimensional space, it is hard to classify OOD inputs into those OOD types that the authors define in the two-dimensional space. In Figure 5, I only see that those projected features are mixed. It is different from what they present in Figure 1.
> > 3. It seems the approach needs to use the test OOD data to train the binary classifier and select the best result, which is not feasible in practice. Usually, it is hard to know what kinds of OOD data we will face. Thus, we should assume that the test OOD data are unknown. I think [1] also uses adversarial examples, which don't depend on the test OOD data, to train the binary classifier and they also show good results. In fact, several recent works start to have this assumption, see [2].
> >
> > [1] Lee, Kimin, et al. "A simple unified framework for detecting out-of-distribution samples and adversarial attacks." Advances in Neural Information Processing Systems. 2018.
> >
> > [2] Hendrycks, Dan, Mantas Mazeika, and Thomas Dietterich. "Deep anomaly detection with outlier exposure." arXiv preprint arXiv:1812.04606 (2018).

---

> > > ### Author Response · Authors · 2020-11-14
> > > **Addressing remaining concerns**
> > >
> > > 1. We respectfully disagree that STL10 and CIFAR10 belong to the same distribution. Would one expect a deep learning model trained on CIFAR10 to generalize and make equally accurate prediction on STL10? But leaving this test-case aside, our experiments also include the standard test-cases used in literature (we also have OOD test-cases with new classes / open world classification in the paper as the reviewer suggested).  We agree that STL10 is similar to the CIFAR10 dataset which makes it more challenging. The low performance (in terms of TNR) of our OOD detector on STL10 with CIFAR10 as in-distribution indicates this challenge.
> > >
> > > 2. They appear mixed because of the projection - but our analysis is not after projection - we do projection just to draw a 2D figure for illustration - our analysis is in the high dimensional space. The sources of uncertainty due to the sample being far from in-distribution data, or far from the class-condition in-distribution data, or close to multiple classes, etc. will remain the same in large dimensions too. Aleatoric or epistemic uncertainty are concepts that are not restricted to smaller dimensions.
> > >
> > > 3.  Reference [1] provided by the reviewer also trains a "logistic regression detector" using OODs (and not adversarial examples) similar to our paper. On Page 5 of Reference [1] ( https://papers.nips.cc/paper/2018/file/abdeb6f575ac5c6676b747bca8d09cc2-Paper.pdf ), the end of the paragraph on "Feature ensemble" states that "In our experiments, following similar strategies in [22], we choose the weight of each layer α` by training a logistic regression detector using validation samples." Reference 22 in that paper is the LID paper from ICLR 2018. So, we are just building on a widely-used notion of OODs.  Reference [2] also does not use adversarial attacks but uses a different dataset for outlier exposure.

---

> > > > ### Comment · AnonReviewer2 · 2020-11-14
> > > > **Concerns still remain**
> > > >
> > > > 1. Yes. STL10 and CIFAR10 don't belong to exactly the same distribution and a model trained on CIFAR10 cannot make an equally accurate prediction on STL10. But we know the distributions of STL10 and CIFAR10 are very close. Since we only have limited training data from CIFAR10 distribution, we cannot model the exact distribution of CIFAR10 distribution. Thus, it is hard to argue that the images from STL10 are not sampled from the CIFAR10 distribution. The model trained on CIFAR10 may be able to give correct predictions on some examples in STL10. Then should we treat these examples as OOD? For example, why should we treat a "ship" image that belongs to CIFAR10 classes and the model could give a correct prediction as OOD? I hope the author could give a **formal definition** of OOD.
> > > >
> > > > 2. Yes. The sources of uncertainty may remain the same in the high-dimensional space. But **the conclusion about which method detects which type of OOD may not hold**. It is hard to argue that those methods fail due to ignoring certain types of OOD in high dimensional space.
> > > >
> > > > 3. Yes. Mahalanobis detector contains a logistic regression detector. But they successfully demonstrate that they can use adversarial examples to train it instead of using test OOD data. My point here is that we cannot use test OOD data to tune the hyper-parameters or train the model since in practice we should **assume that test OOD data are unknown**. I just want to point out that Reference [2] doesn't use test OOD data during training time.

---

> > > > > ### Author Response · Authors · 2020-11-14
> > > > > **Further clarification**
> > > > >
> > > > > 1. The difference in a DNN's performance trained on CIFAR10 on STL10 demonstrates that they are not from the same distribution but as our experiments show they are close to each other. If we filter dataset on which the model makes the correct prediction, we would expect our OOD detection accuracy to improve. The conceptual similarity (same labels) between datasets does not necessarily mean they represent the same statistical distribution. But we completely agree that OODs currently lack a formal definition despite a number of papers on this topic. Our effort in building this taxonomy is exactly motivated by this gap. We hope our paper will be a step towards a better mathematical characterization (and categorization) of OODs.
> > > > >
> > > > > Also, we hope the inclusion of a hard test-case does not penalize our paper because we also included other traditional test-cases.
> > > > >
> > > > > Further, we want to draw the reviewer’s attention to https://www.tensorflow.org/datasets/catalog/stl10, https://cs.stanford.edu/~acoates/stl10/  where it is mentioned that “STL10 dataset is acquired from the labeled examples of Imagenet”. Imagenet is used as OOD set for the CIFAR10 dataset by the state-of-the-art OOD detectors (Mahalanbonis [1], TRAINING CONFIDENCE-CALIBRATED CLASSIFIERS FOR DETECTING OUT-OF-DISTRIBUTION SAMPLES- https://arxiv.org/pdf/1711.09325.pdf,  ENHANCING THE RELIABILITY OF OUT-OF-DISTRIBUTION IMAGE DETECTION IN NEURAL NETWORKS- https://arxiv.org/pdf/1706.02690.pdf etc.).
> > > > >
> > > > > 2. Nothing in our argument about which "OODs depend on what kind of uncertainty (epistemic or aleatoric)" depends on the dimension of data. Our experimental evaluation is over higher-dimensional real datasets. Theoretical study of how high dimensional OODs differ from low dimensional OODs is very interesting, and we agree it would benefit from further investigation. But our identification of taxonomy and experiments with standard benchmarks represent a significant contribution in themselves.
> > > > >
> > > > > 3. We also compared our results with the Mahalanobis technique on adversarial examples. We could beat the Mahalanobis technique on all the tested (FGSM, BIM and DeepFool) attacks for CIFAR10 with ResNet34 model -
> > > > >
> > > > > FGSM    TNR , AUROC, DTACC, AUPR (in), AUPR (out)
> > > > > Mahala - 57.78, 92.87, 85.66, 95.87, 87.52
> > > > > Ours -    61.22, 93.39, 86.61, 96.12, 88.10
> > > > >
> > > > > DeepFool TNR , AUROC, DTACC, AUPR (in), AUPR (out)
> > > > > Mahala -   32.96, 85.80, 78.46, 88.53, 81.68
> > > > > Ours -      41.19, 87.16, 79.51, 88.98, 84.07
> > > > >
> > > > > BIM         TNR , AUROC, DTACC, AUPR (in), AUPR (out)
> > > > > Mahala -  60.70  93.53  86.55  94.61  92.44
> > > > > Ours -      93.23, 98.54, 94.29, 98.39, 98.74
> > > > >
> > > > > Also, please see Section 4.2.2 in Reference 2 ( https://arxiv.org/abs/1812.04606 ) about how they use OOD data. They use filtered Tiny images, ImageNet-22K, etc. as  D_out^OE.  We agree that this outlier exposure is more interesting than using a subset of OOD dataset used in other papers but is of the same nature (not adversarial). We emphasize our notion of OOD is widely used including the references in the review, and we are not the first to use a subset of OOD for tuning our detector.
> > > > >
> > > > > We hope the reviewer will reconsider the score since we have addressed the primary concerns in the review.

---

> > > > > > ### Comment · AnonReviewer2 · 2020-11-14
> > > > > > **Primary concern**
> > > > > >
> > > > > > My primary concern is that the proposed method needs to use a subset of test OOD data for tuning the detector. Although there are some existing works that use the same strategy, recently the community starts to realize that we should assume that the test OOD data are unknown and should not tune detector on a subset of them. I don't ask the authors to provide results on detecting adversarial examples but argue that the Mahalanobis method could achieve good OOD detection performance by tuning the detector on adversarial examples generated on in-distribution examples, which means it doesn't require using the test OOD data to tune its parameters. Unless the authors could demonstrate that their method still works well if they don't tune it on the test OOD data, my concern will still remain.

---

> > > > > > > ### Author Response · Authors · 2020-11-14
> > > > > > > **Addressing primary concern**
> > > > > > >
> > > > > > > We can run our method where we tune the detector with adversarial examples. While most well-cited OOD methods have used the approach similar to our paper (as pointed out in our discussions above which includes the references that the reviewer shared with us), we agree there are alternative ways for tuning. We do not think this is significantly different or challenging for our approach.
> > > > > > >
> > > > > > > Would the reviewer be convinced to improve the score to accept if our results on tuning detector with adversarial examples are as strong as they are currently using a limited amount of outlier exposure?

---

> > > > > > > > ### Author Response · Authors · 2020-11-16
> > > > > > > > **Addressing primary concern with experiments**
> > > > > > > >
> > > > > > > > As per the reviewer's request to demonstrate that our method still works if we don't tune it on the test OOD data, we performed the following two experiments. With ResNet34 trained on CIFAR10, we generated adversarial data (1000 samples) from the test dataset of CIFAR10 for tuning our OOD detector from
> > > > > > > >
> > > > > > > > First Experiment - DeepFool [1] attack.
> > > > > > > > Second Experiment - CW [2] attack.
> > > > > > > >
> > > > > > > > The OOD detector tuned with adversarial examples was then tested with SVHN as OOD at 95% TPR.
> > > > > > > >
> > > > > > > > The results obtained from the DeepFool attack are as follows. TNR    AUROC  DTACC  AUIN   AUOUT-  73.27  94.64  87.52  87.98  97.79
> > > > > > > >
> > > > > > > > The results obtained from the CW attack are as follows. TNR    AUROC  DTACC  AUIN   AUOUT-  59.29  88.08  80.51  70.62  95.22
> > > > > > > >
> > > > > > > > Please note that the OOD detector never saw the tested OODs during its tuning (or training) in the above experiments.
> > > > > > > >
> > > > > > > > [1] Moosavi-Dezfooli, Seyed-Mohsen, Alhussein Fawzi, and Pascal Frossard. "Deepfool: a simple and accurate method to fool deep neural networks." Proceedings of the IEEE conference on computer vision and pattern recognition, 2016.
> > > > > > > >
> > > > > > > > [2] Carlini, Nicholas and Wagner, David. Adversarial examples are not easily detected: Bypassing
> > > > > > > > ten detection methods. In ACM workshop on AISec, 2017.

---

> > > > > > > > > ### Comment · AnonReviewer2 · 2020-11-16
> > > > > > > > > **Need comparison with current state-of-the-art methods**
> > > > > > > > >
> > > > > > > > > Thanks for the experimental results. It seems the results are worse than those reported in the Mahalanobis paper [1]. Could the authors compare their method with current state-of-the-art OOD detection methods, such as Mahalanobis [1], Outlier Exposure [2], and Energy based method [3]?
> > > > > > > > >
> > > > > > > > > [1] Lee, Kimin, et al. "A simple unified framework for detecting out-of-distribution samples and adversarial attacks." Advances in Neural Information Processing Systems. 2018.
> > > > > > > > >
> > > > > > > > > [2] Hendrycks, Dan, Mantas Mazeika, and Thomas Dietterich. "Deep anomaly detection with outlier exposure." arXiv preprint arXiv:1812.04606 (2018).
> > > > > > > > >
> > > > > > > > > [3] Liu, Weitang, et al. "Energy-based Out-of-distribution Detection." Advances in Neural Information Processing Systems 33 (2020).

---

> > > > > > > > > > ### Author Response · Authors · 2020-11-16
> > > > > > > > > > **Comparison with the Mahalanobis and other papers**
> > > > > > > > > >
> > > > > > > > > > We would like to bring to the notice of the reviewer that all the reported results for OOD detection in the Mahalanobis paper [1] are generated with the FGSM attack [2] (right-hand side of table 2 from the paper and its explanation in "Comparison of robustness" on page 7). Our last reported results were with the DeepFool and CW attacks. To make a fair comparison, we ran another experiment with the FGSM attack. Again, the in-distribution data was CIFAR10 trained on the ResNet34 model. The hyperparameters of our OOD detector were tuned only using in-distribution (CIFAR10) and adversarial samples generated by FGSM from the test dataset of CIFAR10. This trained OOD detector was then tested on SVHN as the OOD dataset and here are the results - TNR    AUROC  DTACC  AUIN   AUOUT = 90.18  97.93  92.86  94.11  99.21
> > > > > > > > > >
> > > > > > > > > > The corresponding results reported in the Mahalanobis paper [1] are TNR    AUROC  DTACC - 75.8     95.5    89.1. So, our results are not worse than those reported in the Mahalanobis paper.
> > > > > > > > > >
> > > > > > > > > > For the comparison with the other two papers [3] and [4], here are the results.
> > > > > > > > > >
> > > > > > > > > > With WideResNet trained on CIFAR10 and SVHN as OODs, the results reported in [3] are TNR    AUROC AUPR - 64.41   90.96  97.64 (results from Table 1 in the paper). We used their pre-trained WideResNet (https://github.com/wetliu/energy_ood) on CIFAR10 and trained our OOD detector with the in-distribution CIFAR10 samples and adversarial samples generated by FGSM from the test dataset of CIFAR10.  The OOD detector was then tested on SVHN as OOD and here are the results. TNR    AUROC  DTACC  AUIN   AUOUT- 88.95  97.61  92.46  92.84  99.12.
> > > > > > > > > >
> > > > > > > > > > For comparison with [4], we again used WideResNet trained on CIFAR10 and SVHN as OODs, the results reported in [4] are TNR    AUROC AUPR - 95.2    98.4       89.4 (results from Table 7 in the paper). These results were generating by treating 80M image as OODs for the training of OOD detector. Since this dataset is no longer available, we generated OODs from the TinyImageNet dataset (since Imagenet is OOD for CIFAR10 as well as SVHN) to train our OOD detector. The trained OOD detector was then tested on SVHN as OOD and here are the results. TNR    AUROC  DTACC  AUIN   AUOUT- 92.53  98.56  93.91  96.62  99.41
> > > > > > > > > >
> > > > > > > > > > These experiments were done as per the request made by the reviewer to demonstrate the effectiveness of our method trained without the OOD test data as well as comparison with other recent papers.  Since we have addressed this (major) concern and all the other minor concerns of the reviewer, we would like to request him (or her) to revise his (or her) rating.
> > > > > > > > > >
> > > > > > > > > > [1] Lee, Kimin, et al. "A simple unified framework for detecting out-of-distribution samples and adversarial attacks." Advances in Neural Information Processing Systems. 2018.
> > > > > > > > > >
> > > > > > > > > > [2] Goodfellow, Ian J., Jonathon Shlens, and Christian Szegedy. "Explaining and harnessing adversarial examples." arXiv preprint arXiv:1412.6572 (2014).
> > > > > > > > > >
> > > > > > > > > > [3] Liu, Weitang, et al. "Energy-based Out-of-distribution Detection." Advances in Neural Information Processing Systems 33 (2020).
> > > > > > > > > >
> > > > > > > > > > [4] Hendrycks, Dan, Mantas Mazeika, and Thomas Dietterich. "Deep anomaly detection with outlier exposure." arXiv preprint arXiv:1812.04606 (2018).

---

> > > > > > > > > > > ### Comment · AnonReviewer2 · 2020-11-16
> > > > > > > > > > > **Need the details of the method and experiments**
> > > > > > > > > > >
> > > > > > > > > > > Thanks for the results. I am willing to raise my scores, but I still have some concerns about the experimental details.
> > > > > > > > > > >
> > > > > > > > > > > I think in the current draft, the authors don't describe their approach in detail. For example, the part about how they train a binary classifier is completely missing. Also, the authors mention that they report the best empirical result on the test OOD data out of the 12 ways of combinations of four attributes. I think **we cannot select the best results using the test OOD data**. I hope the authors could describe their approach and experimental setups in detail. It would be good that they could update the current draft to include these details and also new experimental results.
> > > > > > > > > > >
> > > > > > > > > > > Since some details of the approach and the experiments are missing, and the paper needs significant revision, I cannot recommend acceptance for now.

---

### Official Review · AnonReviewer4 · 2020-10-28
**Interesting Work on the Detection of OODs**

**Rating:** 6
**Confidence:** 2

**Review:**

##########################################################################

Summary:

This paper introduces a novel taxonomy for OOD outliers. The authors analyze current OOD detection approaches and uncover their limitations. They propose to fuse several existing approaches into a combined one and extensively evaluate it on various data sets (CIFAR,10, SVNH, MNIST, STL10, ImageNet, etc.). The proposed integrated OOD detection approach clearly shows superior performance.

##########################################################################

Reasons:

Overall, I vote for accepting. The authors make several key contributions: The introduce a novel OOD taxonomy, analyse current OOD detection approaches on a toy data set, propose an integrated OOD detection approach, which shows a superior performance in their extensive evaluation.

##########################################################################

Pros:

* Introduction of a sound and helpful OOD taxonomy
* Limitation analysis of state-of-the-art OOD detection algorithms
* Proposal of a new integrated approach to detect different kind of OOD inputs that unifies the advanatges of underlying algorithms.
* Extensive evaluation of new approach shows clearly superior performance. On a variety of data sets (CIFAR,10, SVNH, MNIST, STL10, ImageNet, etc.) the proposed approach outperforms the baselines on all evaluation criteria (TNR, AUROC, DTACC, AUPR IN, AUPR OUT) for various classifier neural network architectures (LeNet, ResNet, DenseNet).

##########################################################################

Cons:

* The demonstration of the limitations of current OOD detection algorithms is solely empirical (based on a toy data set). Theoretic motivations (if possible) would be a great addition.
* Similarly, a sound theoretical derivation for the proposed integrated approach is lacking.
* Further toy data sets beyond the two half moon data set would be helpful to better understand the implications of all algorithms.

##########################################################################

Questions during rebuttal period:

Please address and clarify the cons above

---

> ### Author Response · Authors · 2020-11-25
> **Addressing comments of the AnnonReviewer4**
>
> We thank the reviewer for the invaluable comments and feedback on this paper. We address the reviewer's comments as follows.
>
> 1. The demonstration of the limitations of current OOD detection algorithms is solely empirical (based on a toy data set). Theoretic motivations (if possible) would be a great addition.
>
> We show the limitations of OOD detection algorithms on the toy dataset for illustration purposes. However, these limitations are applicable to real datasets as well. For example, in Fig 5, we show the examples of the OOD sample caused by epistemic and aleatoric uncertainties on the CIFAR dataset. As mentioned in subsection "Key observations", these samples are missed by the Mahalanobis approach but detected by our approach. This justifies that the definition of ODDs and limitations of the OOD detection algorithms are not limited to toy datasets only.
>
> 2. Similarly, a sound theoretical derivation for the proposed integrated approach is lacking.
>
> The proposed integrated approach is motivated by two key observations: 1) the OOD samples can be of various types (Fig. 1) and 2) one approach cannot detect all types of OODs. (A.2.3 ABLATION STUDY). Thus, an integrated approach is required to detect all types of OODs. Please note the subsection "Key observations" justifies our claims. We agree that a theoretical derivation will further strengthen the claims and we will consider that in the follow-up works.
>
> 3. Further toy data sets beyond the two half-moon data set would be helpful to better understand the implications of all algorithms.
>
> We consider two toy datasets: half moons (Fig 2) and the mixture of Gaussian (Fig 1) to show the variants of OODs. As shown in Fig 2 and 3, after considering a deep neural network to extract features, the non-linear organization of the data points in the original half-moons tends to become linear. This is due to the non-linear projection caused by the neural transformations. We expect to see the same behavior for other 2D or 3D toy datasets as long as a sufficiently deep neural network is considered to extract the features.

---

### Official Review · AnonReviewer3 · 2020-10-30
**Interesting taxonomy of OOD samples but the current paper needs improvement**

**Rating:** 5
**Confidence:** 5

**Review:**

-- Paper Summary:
The paper presents the idea of fusion of attributes from existing sota ood detection methods to achieve higher detection performance.

-- Review :

- The three criteria presented in section two are questions rather than criteria. It is better to be re-worded into criteria.

- Figure 1 suggests the "tied distribution of all training data" is different than the combination of "class distributions". I wish authors  could explain the difference between Type 4 and 5 in the ood sample taxonomy.

- The relation between five types of OOD with three criteria for OOD categorization is not clear.

- The visualization in all figures could be improved:
    - figure 1: too many colors. better to use different shape or numbers directly in the figure.
    - figure 5: not necessary to include, hard to see and comprehend.
    - the total number of figures can be reduced by eliminating some and combining others.

- What was the reason to choose a subset of cifar100 as ood test set but not the whole dataset?

- Authors emphasize reporting detection TNR in the manuscript while FNR is missing from the measurements. I suggest authors either report both or use threshold agnostic metrics like area under precision recall curve (AUPR) or area under receiver operating curve (AUROC) for reporting as in the Table.

- I can't find an explanation and/or discussion on the final detection score and it's hyperparametere.

- The results from the Mahanalobis technique [7] does not match the original paper. If authors did not use a subset of ood samples for tuning, it should be reported in the paper.

-- Strengths:
- interesting taxonomy of ood samples and the following conclusion for integrated detection score.


-- Weaknesses:
- limited on contribution
- no discussion on final detection score and its hyperparameters.
- comparison with more recent techniques including Outlier Exposure [1], Self-supervised reject classifier [2], Geometric self-superivised learning [3,4], and contrastive learning [5,6] are missing in this paper.


[1] Hendrycks, D., Mazeika, M., & Dietterich, T. (2018, September). Deep Anomaly Detection with Outlier Exposure. ICLR 2019

[2] Mohseni, Sina, et al. "Self-Supervised Learning for Generalizable Out-of-Distribution Detection." AAAI. 2020.

[3] Hendrycks, D., Mazeika, M., Kadavath, S., & Song, D. (2019). Using self-supervised learning can improve model robustness and uncertainty. In Advances in Neural Information Processing Systems (pp. 15663-15674).

[4] Golan, Izhak, and Ran El-Yaniv. "Deep anomaly detection using geometric transformations." Advances in Neural Information Processing Systems. 2018.

[5] Tack, J., Mo, S., Jeong, J., & Shin, J. (2020). Csi: Novelty detection via contrastive learning on distributionally shifted instances. arXiv preprint arXiv:2007.08176.

[6] Winkens, J., Bunel, R., Roy, A. G., Stanforth, R., Natarajan, V., Ledsam, J. R., ... & Cemgil, T. (2020). Contrastive training for improved out-of-distribution detection. arXiv preprint arXiv:2007.05566. [8] Liu, Hao, and Pieter Abbeel. "Hybrid discriminative-generative training via contrastive learning." arXiv preprint arXiv:2007.09070 (2020).

[7] Lee, K., Lee, K., Lee, H., & Shin, J. (2018). A simple unified framework for detecting out-of-distribution samples and adversarial attacks. In Advances in Neural Information Processing Systems (pp. 7167-7177).

---

> ### Author Response · Authors · 2020-11-14
> **Addressing comments of AnonReviewer3 [1/2]**
>
> “Figure 1 suggests the tied distribution of all training data is different than the combination of class distributions.”
>
> The tied distribution shown in figure 1 is obtained using robust covariance estimation (https://scikit-learn.org/stable/modules/generated/sklearn.covariance.EllipticEnvelope.html#sklearn.covariance.EllipticEnvelope), representing the unimodal Gaussian distribution of the complete (irrespective of the class) in-distribution dataset and is only meant for illustrative purposes. This is one of the many ways to model the distribution of the complete dataset. Other potential techniques for modeling the complete dataset are kernel density estimation, multi-modal gaussian distribution, etc.
>
> “I wish authors could explain the difference between Type 4 and 5 in the ood sample taxonomy.”
>
> While both Type 4 and Type 5 are OODs due to high class conditional epistemic uncertainty, they vary in their principal component analysis of the class distribution (as explained in the last paragraph of section 2 in the paper). Type 4 are OODs due to high deviation along the principal axis of the in-distribution class 2. Type 5 are OODs due to relatively lower deviation along the non-principal axis (and hence, statistically invariant) of the in-distribution class 1.  The difference in principle axis and non-principal axis will lead to a disparate effect on the reconstruction error.
>
> “The relation between five types of OOD with three criteria for OOD categorization is not clear.”
>
> Criteria 1 - Is the OOD associated with higher epistemic or aleatoric uncertainty, i.e., is the input away from in-distribution data or can it be confused between multiple classes?  OOD Types - Type 1 and Type 2 are OODs due to epistemic uncertainty as they are far from the in-distribution data. Type 3 are OODs due to aleatoric uncertainty between the in-distribution classes 0 and 1.
>
> Criteria 2 - Is the epistemic uncertainty of an OOD sample unconditional or is it conditioned on the class predicted by the DNN model? OOD Types - Type 1 are OODs due to the epistemic uncertainty of the tied in-distribution. In other words, they are far from the tied in-distribution (represented by the red oval in the figure). Type 2 are OODs due to class conditional epistemic uncertainty. In other words, if we consider class-wise instead of a single tied in-distribution then Type 2 are far from all the class distributions. But if we consider the tied distribution, then Type 2 OODs lie within the in-distribution. So, Type 2 are OODs due to class conditional epistemic uncertainty.
>
> Criteria 3 - Is the OOD an outlier due to unusually high deviation in the principal components of the data or due to small deviation in the non-principal (and hence, statistically invariant) components? OOD Types - Type 4 are OODs due to high deviation along the principal axis of the in-distribution class 2. Type 5 are OODs due to relatively lower deviation along the non-principal axis of the in-distribution class 1.
>
> “What was the reason to choose a subset of cifar100 as ood test set but not the whole dataset?”
>
> The subset of CIFAR100 considered in the experiments consists of the following four classes - sea, road, bee, and butterfly. These classes are visually similar to the ship, automobile, and bird classes in the CIFAR10 dataset respectively (as stated in section 4 under CIFAR10 dataset). Therefore, it would make the task of OOD detection with this subset of CIFAR100 for CIFAR10 as in-distribution more challenging. This is the reason for choosing a subset of CIFAR100 to stress test our method on CIFAR10 as in-distribution.
>
> "Authors emphasize reporting detection TNR in the manuscript while FNR is missing from the measurements. I suggest authors either report both or use threshold agnostic metrics like area under precision recall curve (AUPR) or area under receiver operating curve (AUROC) for reporting as in the Table."
>
> AUROC has been reported in Tables 1 and 2 of the experimental section 4  of the paper (comparison with ODIN and Mahalanobis) and Table 3 in the Appendix (comparison with Baseline). AUPR has been reported for all the experiments in Tables 4, 5, 6, 7, 8, and 9 of the Appendix.
>
> “I can't find an explanation and/or discussion on the final detection score and it's hyperparameters”
>
> Section A.2.1 of the Appendix, “Attributes forming the signature of the OOD detector used in the experiments”, explains how the final detection scores are reported.
>
> “Comparison with more recent techniques including Outlier Exposure, … and contrastive learning are missing in this paper.”
>
> We are looking into the suggested papers to compare our results with these papers. We will provide an update on this once we have the results.

---

> > ### Author Response · Authors · 2020-11-14
> > **Addressing comments of AnonReviewer3 [2/2]**
> >
> > “The results from the Mahanalobis technique [7] does not match the original paper. If authors did not use a subset of ood samples for tuning, it should be reported in the paper.”
> >
> > Our OOD detector does not perform input preprocessing (or adding noise to the input) for detection of OODs. So, to compare the results without input-preprocessing, we generated results from the Mahalanobis technique without adding any noise in the inputs.
> >
> > The difference results from Table 1 of the Mahalanobis paper [7] without feature ensemble and without input-preprocessing and our reported results (Table 1- CIFAR10 with ResNet34 on SVHN for TNR, AUROC and DTACC and Table 6 - SVHN on the penultimate layer for AUPR (in) and AUPR (out)) are due to the differences in the floating-point precision of the two machines.
> >                                                                TNR   AUROC  DTACC AUPR(In)  AUPR(Out)
> > Results from Mahalanobis paper - 54.51   93.92    89.13   91.56   95.95
> > Results from our paper -                  53.16   93.85    89.17   91.19   96.14
> >
> > Similarly, the difference in the results from Table 1 of the Mahalanobis paper [4] with feature ensemble and without input-preprocessing and our reported results (Table 2 - CIFAR10 with ResNet34 on SVHN for TNR, AUROC, and DTACC and Table 6 - SVHN on all layers for AUPR (in) and AUPR (out)) are due to the differences in the floating-point precision of the two machines.
> >                                                                TNR   AUROC  DTACC AUPR(In)  AUPR(Out)
> > Results from Mahalanobis paper -  91.45   98.37     93.55   96.43   99.35
> > Results from our paper -                   91.53   98.4       93.63   96.46   99.37
> >
> > The differences in the results from Table 2 of the Mahalanobis paper and our results is because the reported results in Table 2 are obtained after input-preprocessing, whereas we do not consider any input preprocessing for generating results from the Mahalanobis technique.

---

> > > ### Author Response · Authors · 2020-11-17
> > > **Comparison with more recent results**
> > >
> > > For the comparison with the recent papers - "geometric self-supervised learning" [1], "contrastive learning" [2],   "energy-based" [3],  "outlier exposure" [4], we performed the following four experiments (1 experiment per paper).
> > >
> > > 1) With  WideResNet trained on CIFAR10 and LSUN as OODs, the results reported in [1] (table 5) are TNR AUROC DTACC - 71.3 93.2 71.0. With WideResNet as the classifier for CIFAR10, we trained our OOD detector with the in-distribution CIFAR10  as in-distribution samples and adversarial samples generated by FGSM attack [5] from the test dataset of CIFAR10 as OODs. The trained OOD detector was then tested on LSUN as OODs and here are the results. TNR AUROC DTACC AUIN AUOUT- 98.84  99.63  97.72  99.25  99.69.
> > >
> > > 2)  With ResNet-50 trained on CIFAR10 and SVHN as OODs, the results reported in [2] (table 3) are TNR AUROC DTACC AUPR 97.2 99.5  96.7 99.6. With ResNet-50 as the classifier for CIFAR10, we trained our OOD detector with the in-distribution CIFAR10  as in-distribution samples and adversarial samples generated by FGSM from the test dataset of CIFAR10 as OODs. The trained OOD detector was then tested on SVHN as OODs and here are the results. TNR AUROC DTACC AUIN AUOUT- 82.88  96.98  91.74  94.71  98.51
> > >
> > > 3) With WideResNet trained on CIFAR10 and SVHN as OODs, the results reported in [3] are TNR AUROC AUPR - 64.41 90.96 97.64 (results from Table 1 in the paper). With WideResNet as the classifier for CIFAR10, we trained our OOD detector with the in-distribution CIFAR10 as in-distribution samples and adversarial samples generated by FGSM from the test dataset of CIFAR10 as OODs. The trained OOD detector was then tested on SVHN as OOD and here are the results. TNR AUROC DTACC AUIN AUOUT- 88.95 97.61 92.46 92.84 99.12.
> > >
> > > 4) For comparison with [4], we again used WideResNet trained on CIFAR10 and SVHN as OODs. The corresponding results reported in [4] are TNR AUROC AUPR - 95.2 98.4 89.4 (results from Table 7 in the paper). These results were generating by treating 80M image as OODs for the training of OOD detector. Since this dataset is no longer available, we generated OODs from the TinyImageNet dataset (since Imagenet is OOD for CIFAR10 as well as SVHN) to train our OOD detector. The trained OOD detector was then tested on SVHN as OOD and here are the results. TNR AUROC DTACC AUIN AUOUT- 92.53 98.56 93.91 96.62 99.41
> > >
> > > [1] Hendrycks, D., Mazeika, M., Kadavath, S., & Song, D. (2019). Using self-supervised learning can improve model robustness and uncertainty. In Advances in Neural Information Processing Systems (pp. 15663-15674).
> > >
> > > [2] Winkens, J., Bunel, R., Roy, A. G., Stanforth, R., Natarajan, V., Ledsam, J. R., ... & Cemgil, T. (2020). Contrastive training for improved out-of-distribution detection. arXiv preprint arXiv:2007.05566. [8] Liu, Hao, and Pieter Abbeel. "Hybrid discriminative-generative training via contrastive learning." arXiv preprint arXiv:2007.09070 (2020).
> > >
> > > [3] Liu, Weitang, et al. "Energy-based Out-of-distribution Detection." Advances in Neural Information Processing Systems 33 (2020).
> > >
> > > [4] Hendrycks, Dan, Mantas Mazeika, and Thomas Dietterich. "Deep anomaly detection with outlier exposure." arXiv preprint arXiv:1812.04606 (2018).
> > >
> > > [5] Goodfellow, Ian J, Shlens, Jonathon, and Szegedy, Christian. Explaining and harnessing adversarial examples. In ICLR, 2015.

---

> > > > ### Comment · AnonReviewer3 · 2020-11-22
> > > > **Quick reply to authors,**
> > > >
> > > > Thank you for your attention and well-explained response.
> > > >
> > > >
> > > > ----- I still have fundamental issues with the way authors present their OOD types and detection criteria and I encourage authors to stop relying on toy datasets (take the time to read [1] for more on the curse of dimensionality).
> > > >
> > > > - First, you can not separate "the distribution of all training data" from "the combination of class distributions". Using a secondary kernel or other technique to learn training set distribution is not efficient and accurate. The kernel density estimation techniques that you are referring are not as good as (not even close) DNNs when it comes to complex high-dimensional data like images. So, the appropriate way to calculate or visualize the training set distribution is your trained DNN itself. which will tell "the distribution of all training data" is in fact the same as "the combination of class distributions". Note that the model does not learn a perfect representation, which is why uncertainty estimation and OOD detection is still an open problem.
> > > >
> > > > - Second, having the above assumption, it seems to me that you are ended up categorizing different uncertainty types and scenarios that cause model mispredictions errors --- which is valuable ---, rather than a taxonomy of OOD types. Some of these types (e.g. Type 3) that you mentioned could be simply inlier samples which the model did not learn in the train time (i.e. generalization error), see [2] for more example benchmarks.  On the other hand, mispredictions due to samples outside (far or near) the training distribution are "OODs to the trained model" and is a type of distributional error. I think you are pushing to mix these two different model error types. And the outcomes in the flowchart presented in Figure 6, are not guaranteed to be OODs.
> > > >
> > > > ----- I am not convinced about the authors' presentation of results.
> > > > The Mahanalobis technique presented in [7] involves a step for adding noise to the inputs, we can't subjectively disregard steps when implementing other techniques. As a reviewer, I can only compare your results with the MSP baseline.
> > > >
> > > > --- Other 1: Not enough implementation details are presented in the paper, Appendix is for additional info not the only place you present the prediction score. After all, a big chunk of this paper is about how you use multiple detection signals.
> > > >
> > > > --- Other 2: References are not up to date, no 2020 reference, and only 2 references from 2019.
> > > >
> > > >
> > > > [1] Domingos, Pedro. "A few useful things to know about machine learning." Communications of the ACM 55.10 (2012): 78-87.
> > > > [2] Hendrycks, Dan, et al. "Natural adversarial examples." arXiv preprint arXiv:1907.07174 (2019).
> > > > [3] Lee, K., Lee, K., Lee, H., & Shin, J. (2018). A simple unified framework for detecting out-of-distribution samples and adversarial attacks. In Advances in Neural Information Processing Systems (pp. 7167-7177).

---

> > > > > ### Comment · AnonReviewer3 · 2020-11-22
> > > > > **Additional results provided by authors**
> > > > >
> > > > > Thank you for providing additional results in comparison with more recent papers; i. e. geometric self-supervised learning [1], contrastive learning [2], "energy-based" [3], and "outlier exposure" [4]. The goal is to compare and discuss other techniques in your paper to improve it not just throughing a list of numbers. The current manuscript lacks implementation details and discussion of results.

---

> > > > > > ### Author Response · Authors · 2020-11-22
> > > > > > **Addressing comments of AnnoReviewer3**
> > > > > >
> > > > > > We appreciate the invaluable comments provided by the reviewer and provide the following response to address the fundamental issues -
> > > > > >
> > > > > > 1.      We agree with the reviewer’s observation. We consider the toy dataset to illustrate various types of OODs and motivate the need for a taxonomy to capture these variations. We present results only for the benchmark datasets. Note that in Fig. 5, we show the various types of OODs on the features from DNNs for the CIFAR 10. As the reviewer suggested, the distribution is not perfect and the samples belonging to various types of uncertainties are present in the plot. Thus, our definition of taxonomy is not limited to toy dataset only. We will add more such plots based on the DNN feature on the benchmark datasets in the final version.
> > > > > >
> > > > > > 2.      Yes, some of these uncertainties can be considered as modeling errors (e.g., generalization error). However, we categorize the uncertainties (and errors) specifically for OODs and not for general prediction errors. For example, type 3, type 4, and type 5 OODs can all be categorized as the generalization error as they can be mispredicted even though they are close to the data distribution. Thus, a fine-grained categorization, specific to OODs, is needed to analyze various types of OODs.
> > > > > >
> > > > > > 3.      Noise input:  We  (without noise) could outperform (97.07% TNR 99.32 AUROC 96.27 DTACC) the Mahalanobis paper's best-reported result with noise (96.42% TNR 99.14 AUROC 95.75 DTACC) on CIFAR-10 in-distribution data and ResNet34 model.  Our result is reported in table 2 of the paper and Mahalanobis result is reported in Table 1 of the paper.
> > > > > >
> > > > > > 4.      Implementation details: Good suggestion. We kept them in the appendix due to the space limitations but we will add those details to the main draft.
> > > > > >
> > > > > > 5.      Discussion on comparison with recent work: We agree with the reviewer that in addition to the experimental comparison, a discussion is also required for comparison with the latest work. We plan to revise the current draft for including this discussion.

---

> > > > > > > ### Comment · AnonReviewer3 · 2020-11-23
> > > > > > > **Looking forward to seeing the revisions**
> > > > > > >
> > > > > > > Thank you.  I am looking forward to seeing the revision of this paper. Hopefully, the paper is publication-ready by then.
> > > > > > >
> > > > > > > I am expecting some of the reviewers' suggestions to be applied to increase my score. Please maintain a balance between the space used for tables, figures, and the core of the paper. Don't forget that "reviewers are not required" to read appendices, those extra pages are for extra details and lengthy figures/tables. Keep in mind that the research contribution, comparison with similar techniques, and discussions of findings are more valuable than final reported results.

---

### Official Review · AnonReviewer1 · 2020-11-01
**A combination of methods for OOD detection**

**Rating:** 5
**Confidence:** 4

**Review:**

##########################################################################

Summary:
The authors explore the different kinds of outliers and show that the methods previously proposed detect different kinds of OOD and not a single one can detect them all. The authors propose an interesting study of the different kind of outlier on synthetic data which  illustrates well the different characteristics of the outlier types. The authors then propose to combine different methods to increase the OOD detection rate. Experiments are conducted on 3 images classification datasets using different deep neural networks. For each dataset, samples from other databases are introduced as outliers and must be detected. The combination method yield better detection rates than baseline methods in almost all configurations.


##########################################################################

 Reasons for score:

The main idea of the paper is simple : combine different OOD detection metrics to increase the detection rate on different types of outliers. The proposed method indeed increases the OOD detection rate for almost all the experimental settings tested by the authors. However, the method to create the OOD samples is always the same: in-distribution samples come from a database whereas out of distribution samples are drawn from another database. It would be interesting to show that the method also increases the detection rate of outliers inside a given database. This could be done by reporting the classification rate of the DNN in an abstaining scheme : if the OOD metric is greater than a threshold, the sample is not classified (rejected). If the OOD detection method is useful, the classification rate of the DNN can be freely increased by increasing the threshold and rejecting more and more samples.

The author do not justify their choice of the combination method. Computing all the OOD metrics can be computationaly expensive, is it necessary to compute them all ? Are this combination of metric the best ? In which conditions ?

The combination method should be described in the body of the paper, not in appendix.

Guo 2017 appears twice in the bibliography.

---

> ### Author Response · Authors · 2020-11-14
> **Addressing comments of AnonReviewer1**
>
> “show that the method also increases the detection rate of outliers inside a given database .. by reporting the classification rate of the DNN in an abstaining scheme”
>
> As per the reviewer’s suggestion, we conducted new experiments to illustrate the applicability of our approach for the detection of outliers in the same dataset. We considered a ResNet34 model trained on CIFAR10 with an accuracy of 93.67% on the test dataset. We then trained our OOD detector using a set of randomly sampled 300 misclassified images and another set of randomly sampled 300 correctly classified images from the test dataset. This trained OOD detector was then able to correctly identify 326 out of the remaining 333 samples with incorrect predictions as outliers. Thus, using our OOD detector (trained with the True Positive Rate of 95%) with abstaining on outliers improved the classification accuracy to 99.91% from 93.67% on non-abstaining samples. Our results demonstrate that the proposed OOD method can identify and abstain on samples on which the model is likely to produce an incorrect prediction.
>
> “However, the method to create the OOD samples is always the same: in-distribution samples come from a database whereas out of distribution samples are drawn from another database.”
>
> As suggested, we employ a new way to generate out-of-distribution samples by modifying in-distribution samples using adversarial attacks - fast gradient sign method (FGSM) [1], DeepFool [2], basic iterative method (BIM) [3]. We used CIFAR10 as the in-distribution dataset and a ResNet34 model. The attacked images were generated from the test dataset of CIFAR10. We obtained a True Negative Rate (i.e. the rate of detection of attacked images as OODs) of 41.19%, 61.66%,  93.23% for the attacked images generated by Deepfool, FGSM and BIM attacks, respectively at a True Positive Rate of 95%.
>
> The context of the attacks used to generate OODs - FGSM is a single-step attack that uses the L_\infty metric for measuring the distance between a legitimate and perturbed example. BIM was introduced to improve the performance of FGSM by running a finer iterative optimizer for multiple iterations. BIM has a much higher attacking rate than FGSM and it still causes noticeable perturbations even though fewer visual flaws occur than those crafted by FGSM. The perturbations introduced by DeepFool are unnoticeable and the attacking rate is much higher than that of FGSM and BIM.
>
> “Computing all the OOD metrics can be computationally expensive, is it necessary to compute them all ? Are this combination of metric the best ? In which conditions?”
>
> As explained in the Appendix (Section A.2.1) of the paper, the signature of the OOD detector is the weighted sum of the four attributes used to distinguish OODs from the in-distribution samples. These attributes are 1) distance from the in-distribution density estimate, 2) reconstruction error from the principal component analysis, 3) prediction confidence of the classifier, and 4) conformance measure among the nearest neighbors. These attributes can be computed by different metrics. Some of these metrics are mentioned as categories under these attributes in A.2.1. Only one metric per attribute is used in the OOD detector.
>
> As illustrated in Figure 4 of section 3 in the paper, these attributes tend to capture specific types (shown as different clusters) of OODs but not all. Our ablation studies (Tables 10, 11, and 12 of the Appendix) evaluating each attribute show the above-mentioned limitations. Thus, we proposed an integrated approach combining these attributes to detect a diverse type of OODs.  We evaluated our approach on benchmark datasets considering state-of-the-art neural network models.  The proposed approach achieved a better performance as shown in Tables 1 and 2 in the experiment section justifying the importance of the integrated approach.
>
> [1] Goodfellow, Ian J., Jonathon Shlens, and Christian Szegedy. "Explaining and harnessing adversarial examples." arXiv preprint arXiv:1412.6572 (2014).
>
> [2] Moosavi-Dezfooli, Seyed-Mohsen, Alhussein Fawzi, and Pascal Frossard. "Deepfool: a simple and accurate method to fool deep neural networks." Proceedings of the IEEE conference on computer vision and pattern recognition. 2016
>
> [3] Kurakin, Alexey, Goodfellow, Ian, and Bengio, Samy. Adversarial examples in the physical world. arXiv preprint arXiv:1607.02533, 2016.

---

### Author Response · Authors · 2020-11-25
**Addressing major comments of the reviewers in the rebuttal submission**

We greatly thank all the reviewers for their feedback and constructive comments on this work. We really appreciate the time and energy spent by the reviewers on providing their invaluable reviews.

We have tried to address all the major comments by the reviewers in the rebuttal revision -

1) Comparison with the latest (2019, 2020) papers on OOD detection - We have modified our introduction to include the recent papers that were suggested by the reviewers. We have added the "Discussion and Future Work" section for discussion on comparison with these papers. The experimental results for comparison with these papers are reported in Appendix section A.2.1.

2) We have elaborated the process of OOD detection by the proposed OOD detector in the experimental section (section 4) of the main paper.

3) Modification of the experiments for training our OOD detector on adversarial in-distribution samples (and not the test OODs) - The results reported in the experimental section are generated by the proposed OOD detector trained on in-distribution samples and adversarial samples generated from the in-distribution samples as OODs. The previously reported results (performed in supervised settings) have been moved to Appendix (A.2.2).

4) The results generated by the Mahalanobis method in the experimental section are generated from their best settings with feature ensemble and input-preprocessing.

5) We have also reported the metric AUPR in the main experimental section of the paper.

---

### Author Response · Authors · 2020-11-25
**AnonReviewer2**

We thank AnonReviewer2 for agreeing to raise the score. We have also submitted a revised draft that addresses the main concerns. We appreciate the reviewer for the suggestions and a very rewarding discussion.

---

### Author Response · Authors · 2020-11-25
**AnnonReviewer3**

We thank the reviewer for the encouragement. We have submitted a revised draft that addresses the main concerns from the review.

---

### Author Response · Authors · 2020-11-25
**AnnonReviewer4**


We thank the reviewer for the input and we have submitted an improved revised version. Our discussions with other reviewers are also encouraging and we have been able to clarify some major concerns. We hope the reviewer will find the new version to be much stronger and an improved submission.

---

### Author Response · Authors · 2020-11-25
**AnnonReivwer1**

We thank you again for the constructive feedback. We believe your comments raised have been addressed in our response and the updated draft. We'd like to kindly follow up and clarify any remainder confusion. Your feedback has been very important and valuable for us to improve the work!

---

### Decision · Program_Chairs · 2021-01-07
**Final Decision**

**Decision:**

Reject

**Comment:**

**Problem Significance**  This paper introduces an interesting taxonomy of OODs and proposed an integrated approach to detect different types of OODs. The AC agrees on the importance of a fine-grained characterization of outliers given the large OOD uncertainty space.

**Technical contribution** The key idea of the paper is to combine the predictions from multiple existing OOD detection methods. While the AC recognizes the effort made by the authors to address the review comments, reviewers have several major standing concerns regarding limited contributions, insufficient analysis, and lack of clarity. The AC agrees with reviewers that the paper is not ready yet for ICLR publication, and can be further strengthened by:

- (R1) reporting the computational cost for the integrated approach. The inference time for approaches such as Mahalanobis is typically a few times more expensive than the MSP baseline. The cumulative time for calculating all four scores may be non-negligible. Authors are encouraged to analyze the performance tradeoff in a future revision.
- (R2 & R3) discussing the effect of hyper-parameters tuning, which be overly complicated in practice as it involves combinations of multiple methods that each have multiple parameters to tune.
- (R3) comparing with more recent development on OOD detection and move the new results to the main paper. The AC also thinks it's worth discussing the connection and comparison to methods on quantifying uncertainty via Bayesian probabilistic approaches.
- (R2 & R4) more rigorous analysis of the benefits of the proposed integrated approach, both empirically and theoretically. Based on Table 7, the performance of using Mahalanobis alone is almost competitive as the proposed approach (except for the CIFAR10-CIFAR100 pair). This may deem further careful examination to understand what value other components are adding, and in what circumstance.
- (R2, R3 & R4) More discussion on the implication of the taxonomy and algorithm in the high-dimensional space would be valuable. The 2D toy dataset might be too simple to reflect the decision boundary as well as uncertainty space learned by NNs. Moreover, it's important to justify further how aleatoric and epistemic uncertainty manifests in the current experiments using NNs. For example, epistemic uncertainty can arise due to the use of limited samples or due to the model uncertainty associated with the model regularization.

Recent work by Hsu et al. [2] also attempt to define a taxonomy of OOD inputs (based on semantic shift and domain shift), which can be relevant for the authors.

**Recommendation** Three knowledgeable reviewers have indicated rejection. The AC discounted R4's rating due to the less familiarity in this area and lack of participation in the post-rebuttal discussion.

[1] Richard Harang, Ethan M. Rudd. Towards Principled Uncertainty Estimation for Deep Neural Networks
[2] Hsu et al. Generalized ODIN: Detecting Out-of-distribution Image without Learning from Out-of-distribution Data